# Global resource potential of seasonal pumped hydropower storage for energy and water storage

Julian D. Hunt [1✉], Edward Byers [1], Yoshihide Wada [1], Simon Parkinson [1,2], David E.H.J. Gernaat[3,4], Simon Langan [1], Detlef P. van Vuuren [3,4] & Keywan Riahi[1]

Seasonal mismatches between electricity supply and demand is increasing due to expanded use of wind, solar and hydropower resources, which in turn raises the interest on low-cost seasonal energy storage options. Seasonal pumped hydropower storage (SPHS) can provide long-term energy storage at a relatively low-cost and co-benefits in the form of freshwater storage capacity. We present the first estimate of the global assessment of SPHS potential, using a novel plant-siting methodology based on high-resolution topographical and hydrological data. Here we show that SPHS costs vary from 0.007 to 0.2 US$ m$^{-1}$ of water stored, 1.8 to 50 US$ MWh$^{-1}$ of energy stored and 370 to 600 US$ kW$^{-1}$ of installed power generation. This potential is unevenly distributed with mountainous regions demonstrating significantly more potential. The estimated world energy storage capacity below a cost of 50 US$ MWh$^{-1}$ is 17.3 PWh, approximately 79% of the world electricity consumption in 2017.

[1] International Institute of Applied Systems Analysis (IIASA), Laxenburg, Austria. [2] Institute for Integrated Energy Systems, University of Victoria, PO Box 3055 STN CSC Victoria, BC, Canada. [3] Copernicus Institute of Sustainable Development, Utrecht University, Heidelberglaan 2, 3584 CS Utrecht, The Netherlands. [4] PBL – Netherlands Environmental Assessment Agency, Bezuidenhoutseweg 30, 2594 AV Den Haag, The Netherlands. ✉email: hunt@iiasa.ac.at

Whilst a number of energy storage technologies are being developed to manage electricity grids, most technologies only fulfil short-term cycles (daily or shorter). Pumped hydropower storage (PHS) systems are currently the most mature and widespread method for large-scale electricity storage[1–6]. Global installed PHS electricity generation capacity is ~165 GW and constitutes the vast majority of electricity storage worldwide, of which 25 GW have been identified as mixed plants that are also conventional reservoir-based hydropower dams[7]. Often, PHS is seen as a technology capable of storing energy for daily or weekly cycles and up to months[8–13]; however, the technology can also operate over annual and pluriannual cycles[14]. Given the current costs reduction in other technologies offering daily energy storage (particularly batteries), PHS is anticipated to gain importance as a seasonal energy and water storage alternative.

A SPHS plant consists of a high-head variation storage reservoir built in parallel to a major river. During periods of low-energy demand or high water availability, water is pumped into the reservoir. Stored water is released from the reservoir generating electricity when additional electricity generation capacity is required, or water is scarce. They can be compared to conventional reservoir dams, due to the possibility of regulating the river flow and increasing the hydropower generation on the hydropower dams in cascade[15]. SPHS plants have lower land requirements than conventional hydropower dams, for a comparable energy and water storage potential, because the off-river reservoir design permits higher hydraulic heads variations[14]. SPHS can also be attractive to deal with the load problems emerging from electricity consumption and supply seasonal variations and increasing use of intermittent sources of generation. The storage of water can also help to overcome water shortage problems. Because storage is also not near the main river, possible negative impacts of hydropower can be better managed (further details in Supplementary Table 1).

To understand the potential that SPHS can fulfil in future energy storage requirements, in this paper, we present the first comprehensive and globally consistent assessment of SPHS potential. It presents the results from the SPHS world potential model, which is an upgrade of the methods that have been used for estimating global hydropower potential[16–21]. One recent study investigates the global potential for PHS and assumes the construction of two reservoirs in a closed loop for daily and weekly operation. They found a global potential of $23 \times 10^6$ GWh in more than 600,000 plants, but the project sizes appear to be impractical or infeasible for seasonal storage or water storage and do not include detailed cost analysis or water availability[22,23] (Supplementary Table 2). We have not included these closed loop sites because they are designed to store energy and we are looking at energy and water storage solutions in this paper. Other studies have been developed to find the potential for PHS projects in Europe[21,24,25], and Iran[26], however, these are regional models also do not include costs. In this paper, we scan the global landscape alongside rivers for attractive sites to build artificial reservoirs for water and energy storage purposes with SPHS plants. Here, we evaluate all land grid points for project suitability at a 15 s resolution (~450 m resolution), using a detailed siting assessment methodology for developing and costing SPHS projects with topography, river network, and hydrological data.

Our estimates show that the global technical and economic potential for water and energy storage with SPHS is vast, but with an unequal spatial distribution across the world. Considering all the energy storage projects with the cascade, the total storage capacity is equivalent to 17,325 TWh, or ~79% of the world electricity consumption in 2017. Whilst we have considered a maximum of one SPHS per 1-degree grid square (100 × 100 km),

in some locations a series of SPHS plants in cascade could further increase the energy storage potential.

## Results

The SPHS world potential model identified more than 5.1 million potential projects, all of which have a fixed generation/pumping capacity of 1 GW. With the intention of eliminating competing projects and focusing on the best projects per region, the projects with the lowest costs for water storage (US\$ m$^{-3}$), long (US\$ MWh$^{-1}$) and short-term (US\$ kW$^{-1}$) energy storage, within a 1 arc degree resolution of the globe are presented. This consists of 1457 water storage projects with water storage costs lower than 0.2 US\$ m$^{-3}$ and 1092 energy storage projects with energy storage cost lower than 50 US\$ MWh$^{-1}$ (some of the water projects consist of the same energy projects).

Critical components of the SPHS project costs are the dam and tunnel (Fig. 1a). Tunnel costs increase proportionally with its length and reduce with generation head. Dam costs increase proportionally with width and exponentially with height. A high land-value of 41,000 US\$ ha$^{-1}$ was assumed in this paper, and it represents typically 5% of the total project costs. It is important to emphasize the relatively low land requirement of SPHS in comparison to conventional hydropower dams that have smaller variations of reservoir levels and thus flood more area for the same water storage capacity. The average level variation of SPHS projects is 151.7 m for energy storage projects (Supplementary Table 5).

We use a site in Tibet, China to illustrate the calculations (Fig. 1b, c). With a 50 m dam height, the energy storage costs are the highest at 11.7 US\$ MWh$^{-1}$. Most of the costs are related to the tunnel costs (45%), which is 18 km long. The land cost is high (8%) if compared to the dam costs (7%) because the amount of water stored per km$^2$ is low. Energy storage cost is the lowest for a 150 m dam height. In this case, the tunnel cost is 30%, and dam costs, 36% and land cost is low (6%) (Supplementary Tables 6 and 7). A further increment in head increases energy storage costs, mostly because the required water to fill up the reservoir according to Eq. (3) exceeds the maximum flow extraction from the river.

Looking at the global potential, the water storage cost with SPHS varies from 0.007 to 0.2 US\$ m$^{-3}$ of water stored (Fig. 2a). This large cost difference is due to the variation in topography and water availability. The energy storage cost varies from 4.6 to 50 US\$ MWh$^{-1}$ without including dams in cascade and from 1.8 to 50 US\$ MWh$^{-1}$ when including them (Fig. 2b, c, respectively). The water stored in a SPHS plant also benefits the dams downstream (in cascade). The higher the altitude of the SPHS system, the more energy it stores for the whole basin. Given that the SPHS projects proposed in this paper intend to regulate the flow of the river, if the river has dams downstream the SPHS plant, they will also generate more electricity with the flow released from the SPHS plant[15]. Assuming a cost for natural gas storage of 1 US\$ mcf$^{-1}$ [27] and an electricity generation efficiency of 50%, the cost of energy storage with natural gas is ~6.8 US\$ MWh$^{-1}$. This value is higher than the energy storage with SPHS in mountainous regions with cascade around the world (Fig. 2c). The world storage capacity curves are shaded because they include the cheapest projects and a combination of cheap and large storage capacity projects.

The cost of 1 GW PHS capacity varies from 370 to 600 US\$ kW$^{-1}$ (Fig. 2d). This excludes dam and land costs. The costs are segmented in different steps due to the variation in length of the tunnel, which starts at 3 km with additional increments of 3 km. A cost comparison of other short-term energy storage technologies can be found in ref. [28].

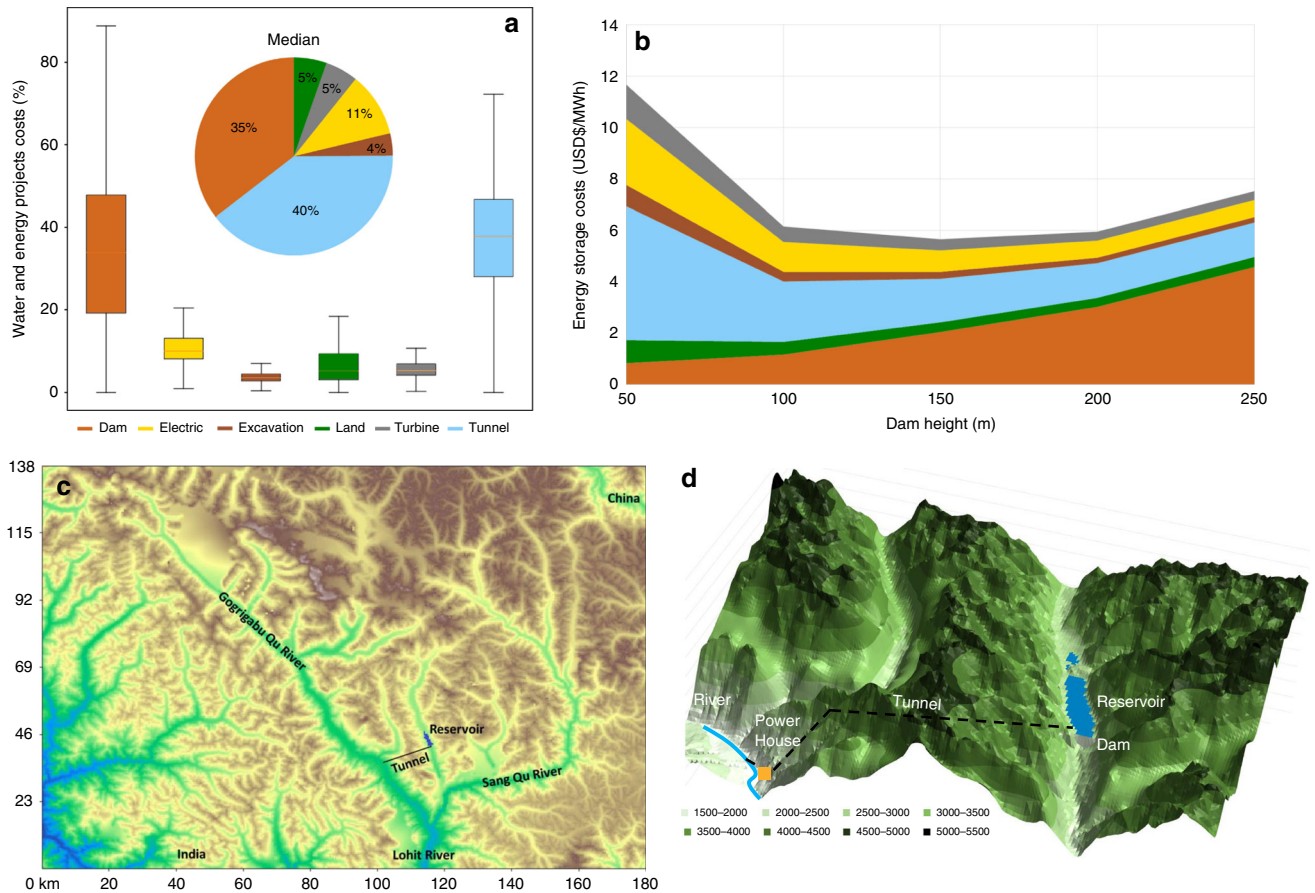

**Fig. 1 Seasonal pumped hydropower storage (SPHS) costs and description. a** Water and energy SPHS project cost distribution shows that the most expensive components tend to be the tunnel and dam. **b** Example of energy storage cost variation with cascade according to different heights for the example project in **c**. The energy storage cost reduces with the increase in dam height due to economies of gains, however, it then increases because the reservoir becomes larger than the amount of water available to be sustainably stored. **c** Presentation of selected project in Tibet, China, on a topographic map, presenting its tunnel in black and reservoir in purple. **d** Zoom in the selected project.

The percentage of inflow from the tributary river to fill up the reservoir varies for each project (Fig. 2e). The remaining percentage consists of the water that is pumped into the SPHS reservoir from the river below. Three of the proposed projects, over more than 1000 projects, have 100% of the inflow coming from the tributary river. In these cases, a reversible turbine would be interesting only to allow the project to store energy in daily and weekly cycles, given that the seasonal cycle is already accomplished with the river flow.

The land requirement for energy storage varies from 1.2 to 20 $km^2\,TWh^{-1}$ (Fig. 2f). This is a result of the high water level variation in SPHS reservoirs (mean 141 m for water storage and 152 m for energy storage (Supplementary Table 5)). For comparison, the average land requirement for hydropower energy storage in Brazil is around 150 $km^2\,TWh^{-1}$ [29]. The low land requirement of SPHS projects makes it a more social and environmentally friendly storage alternative when compared with conventional dams. Large reservoirs with longer storage cycles usually have lower storage costs than small ones. The storage cycle will depend on the needs for storage and the storage potential of the reservoir.

## Discussion

Conventional hydropower dams have been built in main river channels with the intention of managing water resources and generating low-cost, low-carbon electricity, but often they fragment flow and flood upstream areas. SPHS plants built adjacent to main rivers can provide similar water management and energy storage services while avoiding the large land footprint associated with conventional hydropower dams. This paper has identified where SPHS plants could be built and the associated unit costs for energy and water storage services. The estimated potential is restricted to mountainous regions with reasonable water availability and high hydraulic heads supporting cost-efficient SPHS system design. Significant potential exists in the lower part of the Himalayas, Andes, Alps, Rocky Mountains, Northern part of the Middle East, Ethiopian Highlands, Brazilian Highlands, Central America, East Asia, Papua New Guinea, the Sayan, Yablonoi and Stanovoy mountain ranges in Russia, with energy storage costs with cascade varying from 1.8 to 50 $US\$\,MWh^{-1}$ (Fig. 2c).

SPHS projects are shown to provide multiple income generating services, for example, a single SPHS project provides water storage at 0.1 $US\$\,m^{-3}$, long-term energy storage at 30 $US\$\,MWh^{-1}$ and short-term energy storage at 600 $US\$\,kW^{-1}$. Considering that the need for three storage services are complementary in the SPHS projects, the costs of these services are substantially reduced [14]. The change in cost for each storage service will vary with the need for storage and the operation of the SPHS plant. Compared with natural gas storage, this work has shown that there is considerable potential for SPHS to provide competitive storage, noting that the gas comparison does not even consider the cost of the gas power plant so as not to confuse storage with generation.

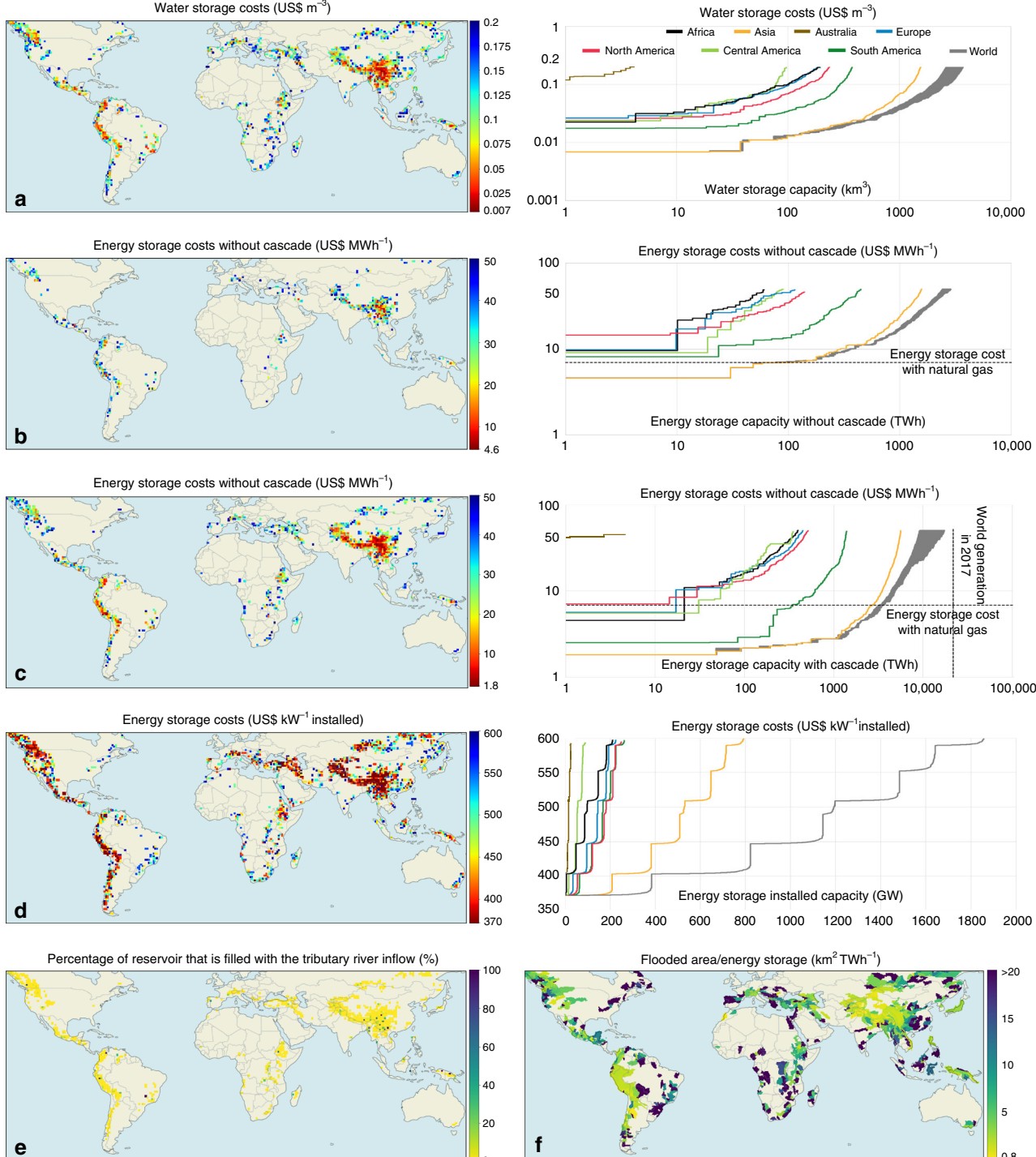

**Fig. 2 Seasonal pumped hydropower storage world cost and flooded area maps. a** Water storage costs and capacity curve in km³. **b** Energy storage without considering hydropower plants in cascade costs and capacity curve in US$ MWh⁻¹. **c** Energy storage considering hydropower plants in cascade costs and capacity curve in US$ MWh⁻¹. **d** Additional generation capacity costs and capacity curve in US$ kW⁻¹. **e** Percentage of the reservoir that is filled with the river inflow into the SPHS reservoir. **f** Average land requirement for energy storage in different basins.

The needs for energy and water storage with SPHS plants should be complementary. This is because during the dry season there will be low volumes of water available to be used for energy storage. This complementarity is usually the case in high latitude countries, where during the summer river flow is higher due to ice melting and energy demand is lower compared to the winter. Inter-tropical regions with abundant hydropower generation also

have complementarity, where during the wet season there is high water availability and hydropower generation. However, there are regions and countries where the need for energy and water storage is not complementary, for example, in the inter-tropical regions without hydropower generation, where the summer and wet season is the period with highest electricity demand due to air conditioning. In cases where energy and water storage need are

not complementary, SPHS should not be considered as an energy and water storage alternative.

Given that this is the first global assessment for SPHS, the model was developed with the intent of focusing on its technical potential. Other restrictions that impact socio-economic feasibility, such as population, land use, biodiversity, transmission, etc. were not included in this work with the intent of presenting the existing potential and not its viability. Regional studies such as, Rogeau et al. [21] already tried to eliminate irrelevant sites due to conflicts with existing land use.

With the needs for reducing $CO_2$ emissions to mitigate the impacts of climate change, SPHS provides short-term and long-term energy storage services allowing the development of 100% renewable energy grids. SPHS also increases water security in regions with unsuitable topography for conventional dams, high evaporation, and sedimentation rates. It is, thus, a prominent alternative for sustainable development on a worldwide scale.

## Methods

**Global model.** To assess the global potential of SPHS, our methodology integrates five critical components, which are: topography, river network and hydrology data, infrastructure cost estimation and project design optimization. SPHS project suitability mainly depends on the topography, distance to a river and water availability, which together determine the technical potential. Additional contextual factors, such as distance from energy demand and associated transmission infrastructure losses and associated costs, determine the economic feasibility. Whilst previous studies have used similarly high-resolution topography for conventional reservoir dams estimation[16], the possibility of storing water and energy by pumping water to a reservoir parallel to a major river has not been globally assessed. Since storage potential and infrastructure costs are highly dependent on the topography, our new spatially explicit approach identifies numerous technically feasible candidate sites and provides estimates of costs.

Details of the SPHS world potential model framework are explained step-by-step in Fig. 3 and Supplementary Tables 3–5. The model goes through each grid cell location delineated at a 15″ resolution, implementing a detailed siting assessment that accounts for topography and hydrology in the calculation of project-level costs. The model performs the stages as follows. First, it looks for a river with reasonable flowrate up to 30 km away from a reservoir (Fig. 3c), second it checks if a dam up to 250 m high can be built from the grid cell (Fig. 3d), third it removes projects with competing dams (Fig. 3e), fourth it finds the flooded side of the dam and creates the reservoir (Fig. 3f), fifth it calculates the volume and flooded areas, sixth it compares the size of the storage site with the water available for storage (Fig. 3g), seventh it estimates the costs of the dam, tunnel, turbine, generator, excavation and land, eighth it estimates water and energy storage costs (Fig. 3i).

SPHS reservoirs in this study are operated to reduce the seasonality and interannual river flow variations, to regulate the flow of the river. If the river flow is already constant, then the water available for storage in the SPHS reservoir will be zero, as it would deregulate the flow of the river. The hydrological data were used to restrict the size of the storage reservoirs, according to water availability. This guarantees that there will be water available to fill up the storage reservoir without having a considerable impact on the overall river flow. Additionally, conservative storage values are assumed to reduce the impact of the SPHS plant in the river flow. The maximum volume of the reservoir equals to 11% of the annual river flow, from which the need for storage is divided by seasonal storage needs and inter-annual storage needs. This value was selected with the intent of reducing the environmental impact of storage on the overall river flow. On average, the water available for storage in SPHS reservoirs in this model equals to 7.7% of the annual river flow (Supplementary Fig. 1).

**Data collection.** The SPHS World Potential Model provides a near-global scale potential (56°S–60°N). The excluded area is due to unavailable Shuttle Radar Topography Mission (SRTM) topographic data. The topographic data applied is SRTM[30] and has 3″ resolution. The resolution is decreased to 15″, assuming the centre point, to reduce modelling time and to combine with the river network data. The river network data assumed the Strahler methodology in Global-scale river network (GRIN)[31], which is derived from the SRTM data and has 15″ resolution. The topographic and river Strahler data are then combined with the hydrological data taken from the PCR-GLOBWB[32] global hydrological model, which are derived into annual discharge, seasonal, and inter-annual variations. We use methods to optimize the number and diameter of the tunnels[33] and for cost-estimation[34] procedures, which are further explained in Supplementary Tables 2 and 3.

**Site selection model and engineering design.** The site selection model is divided into nine main stages (Supplementary Table 4). Initially, the topographic

data are combined with the river Strahler data, which is a numerical measure of its branching complexity and was derived from the same topographic data. The higher the Strahler stream order value the more tributary rivers (branches) deliver water to the given part of the river[31]. The two are compared at the same resolution to identify rivers within the topographic data and to estimate the length of the tunnels connecting the upper reservoir and the lower reservoir in the river. The river Strahler data are also used to reduce the amount of SPHS projects developed within a given area and reduce the modelling time. Rivers with small river Strahler stream order have low river flows, which results in unviable SPHS projects with small generation capacities. A minimum river Strahler stream order of 7 is selected because the river has a considerable number of tributary rivers connected to it, which results in a relatively large and constant river flow.

For each land grid cell at a latitude between 60° N and 56° S (point under analysis (PUA)), the model searches for rivers with sufficient discharge (river Strahler stream order ≥7) within 1–30 km of distance, which consists of the tunnel length. If a large river is found, the model attempts to build dams of 50, 100, 150, 200 and 250 m height, along four axes (N–S, W–E, NW–SE, NE–SW) and with a maximum length of 7.2 km (Supplementary Table 4 (L4)). If the topography allows the construction of such dams, it verifies whether the PUA is the lowest point of the dam (if it is not, the process stops with the intention of not repeating the same project). Using the surrounding topography and observing limits to the maximum flooded area of the reservoir, the model identifies the side with the largest storage volume. Subsequently, the reservoir water level is varied to determine the flooded area vs. level and storage volume vs. level curves. This is done by subtracting the volume of land and water with the reservoir at a given level by the volume of land and water with the reservoir at its minimum level. Project costs are estimated using the equations presented in the ref. [34]. The cost is divided in dam, tunnel, powerhouse excavation, pump-turbine, electro-technical equipment and land costs. More details on the assumptions for the cost estimate are presented in Supplementary Table 3. In the analysis, the water storage capacity of the SPHS projects is limited according to the water availability of the main river. The maximum water storage capacity is limited to 11% of the annual river flow, which is a small portion of the river flow and results in a small impact to the river. If the storage capacity is much higher than the amount of water available, the estimated cost of storage increases, as a section of the reservoir will never fill up. The project costs are then compared with the hydrology of the river to find the water and energy storage costs with Eqs. (4) and (5).

**Storage dimensioning and cost.** The seasonal and interannual variability of river discharge used in the Hydrological Analysis Stage is calculated with the Eqs. (1) and (2). They are important to calculate the water available for storage in the SPHS reservoir, with the objective of producing a constant river flow. If the river has no seasonal variation, then the water available for storage would be equal to zero. This is because, if the SPHS deregulates the flow of the river, the hydropower potential of the dams in cascade or the water supply downstream could be negatively affected.

$$S_V = \frac{\sqrt{\frac{\sum_{m \in M}(\bar{q}_m - \bar{q}_s)^2}{N_m}}}{\bar{q}_s} \quad \{\text{if } S_V > 1 \rightarrow S_V = 1\} \tag{1}$$

$$I_V = \frac{\sqrt{\frac{\sum_{y \in Y}(\bar{q}_y - \bar{q}_s)^2}{N_y}}}{\bar{q}_s} \quad \{\text{if } I_V > 1 \rightarrow I_V = 1\} \tag{2}$$

where, $S_V$ is the seasonal variation index, $I_V$ is the interannual variation index, $\bar{q}_m$ is the river flow of a given m month, $\bar{q}_y$ is the average river flow of a given year, $\bar{q}_s$ is the average river flow over y years, $N_m$ is the number of months, $N_y$ is the number of years.

The costs of water and energy storage services calculated in the Estimate Storage Cost stage vary according to the annual river flow, the seasonal and interannual variation indexes. These hydrological parameters have three main purposes. Firstly, they intend to guarantee that there will be sufficient water in the river to be stored in the upper reservoir. Secondly, the need for water and energy storage should not have a substantial detrimental impact on the river flow. Thirdly, the water storage potential intends to regulate the flow of the river and produce a constant flow of water, reducing its seasonality and interannual variations.

Using the values calculated for $S_V$ and $I_V$ (Supplementary Fig. 2) and the percentage of river annual discharge available for storage (Supplementary Fig. 1), the water available for storage $Q_A$ (Eq. (3)) is calculated by

$$Q_A = Q \times (S_V \times 0.1) \times (1 + (I_V \times 0.1)) \tag{3}$$

where $Q_A$ is the water available for storage in $km^3 \, yr^{-1}$, $Q$ is the river annual discharge in $km^3 \, yr^{-1}$.

The variation of the water and long-term energy storage costs with the water available for storage is presented in Eq. (4). The costs for additional short-term energy storage are presented in Eq. (5). For more details on Eqs. (1)–(5) please refer

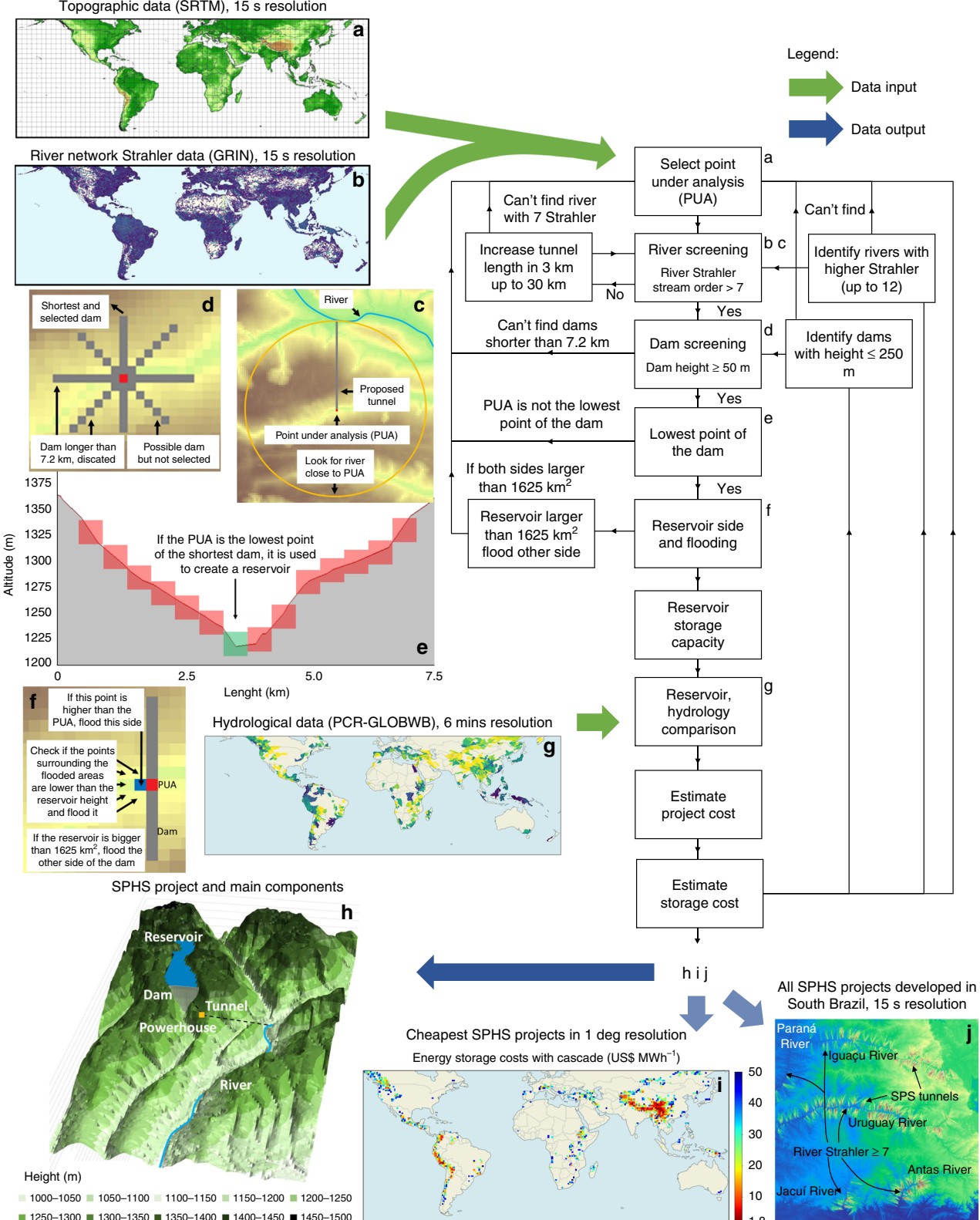

**Fig. 3 Seasonal pumped hydropower storage (SPHS) world potential model framework. a** Topographical data input from the Shuttle Radar Topography Mission (SRTM)[30]. **b** River network Strahler data input from the Global River Network (GRIN)[31]. **c** Finding rivers close to the SPHS site. **d** Looking for possible dams. **e** Limiting the number of proposed SPHS projects. **f** Creating and finding reservoirs. **g** Hydrological data input[32]. **h** Representation of a possible SPHS project in the Zambezi river basin. **i** Cheapest SPHS projects in 1-degree resolution. **j** Location with several SPHS projects proposed.

to the Supplementary Table 2.

$$C_{\mathrm{W}} = \frac{C_{\mathrm{P}}}{W_{\mathrm{S}}}; \; C_{\mathrm{Ewc}} = \frac{C_{\mathrm{P}}}{E_{\mathrm{Rwc}}\frac{W_{\mathrm{R}}}{W_{\mathrm{S}}}}; \; C_{\mathrm{Ewoc}} = \frac{C_{\mathrm{P}}}{E_{\mathrm{Rwoc}}\frac{W_{\mathrm{R}}}{W_{\mathrm{S}}}} \begin{cases} \text{if } W_{\mathrm{R}} < Q_{\mathrm{A}} \rightarrow W_{\mathrm{S}} = W_{\mathrm{R}} \\ \text{if } Q_{\mathrm{A}} < W_{\mathrm{R}} < 2Q_{\mathrm{A}} \rightarrow W_{\mathrm{S}} = Q_{\mathrm{A}} + 0.5W_{\mathrm{R}} \\ \text{if } W_{\mathrm{R}} > 2Q_{\mathrm{A}} \rightarrow W_{\mathrm{S}} = 1.5Q_{\mathrm{A}} \end{cases}$$

(4)

where $C_{\mathrm{W}}$ is the cost of water storage in US\$ km$^{-3}$, $C_{\mathrm{P}}$ is the cost of the project (i.e. dam, tunnel, turbine, electrical equipment, excavation, and land) in US\$, $W_{\mathrm{s}}$ is the water storage capacity adjusted by the water availability in km$^3$, $C_{\mathrm{Ewoc}}$ is the cost of long-term energy storage excluding the cascade in US\$ MWh$^{-1}$, $C_{\mathrm{Ewc}}$ is the cost of long-term energy storage including the cascade in US\$ MWh$^{-1}$, $W_{\mathrm{R}}$ is the water storage capacity of the reservoir developed in the model in km$^3$, $E_{\mathrm{Rwc}}$ and $E_{\mathrm{Rwoc}}$ are the energy storage capacity of the reservoir developed in the model with and without cascade in MWh, respectively.

$$C_{\mathrm{GW}} = \frac{C_{\mathrm{PGW}}}{G}$$

(5)

where, $C_{\mathrm{GW}}$ is the cost of additional generation capacity in US\$ kW$^{-1}$, $C_{\mathrm{PGW}}$ is the cost of additional generation capacity (i.e. tunnel, turbine, electrical equipment, excavation) in billion US\$, $G$ is the generation capacity in GW (fixed to be 1 GW for all SPHS plants proposed).

**Reporting summary**. Further information on research design is available in the Nature Research Reporting Summary linked to this article.

## Data availability
The data that support the plots within this paper and other findings of this study are available from the corresponding author upon reasonable request.

## Code availability
The code is available from the corresponding author upon reasonable request.

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

## Acknowledgements
We would like to thank CAPES/BRAZIL for the research grant as part of the CAPES/IIASA Postdoctoral Programme.

## Author contributions
J.D.H. conceived the idea, developed the modelling techniques and led the manuscript writing, E.B. contributed to modelling and to the concept, Y.W. and D.E.H.J.G. developed the hydrological datasets used in the paper, S.P. contributed to the model and references, S.L. contributed to the water availability restriction to the model and references, D.P.V. and K.R. perfected the idea with valuable inputs. All authors contributed to the manuscript.
