## [Peer Review File · Nature Communications]

Reviewers' comments:

Reviewer #1 (Remarks to the Author):

KEY RESULTS

This work introduces the first high-resolution assessment of seasonal pumped hydro energy storage potential at global scale.

Pumped-storage plants are used to store in a single project large amounts of water and energy in off-stream reservoirs during periods of high river flows or excess energy in the grid and to generate electricity by releasing water at times of highest system power demand. Pumped hydro energy storage is a mature and available technology, which has existed for a long time but is scarcely used on a seasonal fashion. Seasonal pumped-storage allows long-term storage of seasonal surplus energy, adding operating flexibility to the system while providing water flow regulation service at the same time.

The model developed in this work identifies more than 1,000 suitable locations worldwide using a systematic approach, based on high resolution topographical data and discharge data from a global hydrological model. It also predicts the associated costs for water, short and long-term energy storage services.

This study shows that there is a considerable potential for seasonal pumped-storage to provide competitive storage, although this potential varies substantially from region to region, being restricted to mountainous regions.

ORIGINALITY AND SIGNIFICANCE

The results presented in this work are **of strong interest to the international community**, regardless of discipline because of their social implications. As pumped storage hydropower is one of the most promising technologies for balancing seasonal fluctuations of intermittent renewables, identification of technically feasible sites for new installations and improved information on their costs are **crucial**. This work represents an advance in assessing global capacity for energy storage, likely to influence power system planners and foster the deployment of pumped hydro energy storage systems.

The world potential of hydropower, which is currently the dominant renewable energy source, has already been evaluated. Previous studies also exist that quantify pumped hydro storage capabilities at regional scales, such as in Europe. As an attempt to comprehensively assess the world potential of seasonal pumped-storage using an upgraded model synthesising previous approaches, the results presented in this work are **original**. Also **novel** is the associated cost analysis presented here, despite the difficulties to provide an accurate global estimate of the costs.

The authors nonetheless briefly mention the global atlas of pumped hydro energy storage (Ref. 18). In their report, Stocks et al. from the Australian National University found a total of more than 600,000 potentially feasible pumped hydro energy storage sites worldwide with a massive storage potential of about 23 million GWh. The identified sites are mostly closed-loop, comprising an upper and lower reservoir pair, which is ranked according to an approximate cost model.

This very recently released database cannot simply be swept aside in a single sentence (L50-52) but need further discussion. One can wonder, for instance, to what extent the close-loop sites identified in the global pumped hydro atlas are complementary to or coincide with the open-loops considered in this study (Supplementary Table 7, L15C2). This also points to a need for a clearer

definition of a seasonal pumped-storage project in the introduction (L34-37, see Ref. 10). In addition to be helpful for those unfamiliar with the technology, it would help clarify its specificities. In particular, it is worth mentioning from the beginning that:

- "the river can have a small reservoir" (Supplementary Table 7, L15C2),
- the energy storage potential also includes the energy generated at the dams in cascade downstream the plant (L137-139 & Supplementary Table 7, L2),
- upper "sites are not restricted to rivers" (L198) but usually coincide with a tributary river feeding directly the reservoir (Supplementary Table 3, L5C2).

CLARITY AND CONTEXT

The abstract is clear and accessible. It focuses attention on seasonal pumped-storage costs but could also mention the total number of selected projects with the lowest costs (L103-108) and point out the unequal spatial distribution of the considerable evaluated capacity (L17-18, L181-182).

The authors should also state in the abstract that they present a **rough** (Supplementary Table 7, L9C2) global assessment of seasonal pumped-storage **theoretical** potential, "focusing on its technical potential" (L204-207) and without human considerations and climate change. Indeed viability of such projects remains a major challenge, which somewhat limits the scope of this study. Despite the benefits of deploying seasonal pumped-storage (Supplementary Table 1), any new infrastructure will face conflicts with existing land use and significant costs might occur from mitigation measures. One might therefore ask to what extent these "restrictions that impact socio-economic feasibility" would influence the estimated costs and energy storage capacity, all the more since a single world capacity value is provided.

VALIDITY, DATA & METHODOLOGY

The description of the first model stages (1 to 7) in the main body of text, in Fig. 2, in the Methods section and in the Supplementary Tables 2 and 3 is quite **redundant**, without a lot of added value. The text should be more concise regarding these stages but more explicit in reporting cost estimation methodology, which is not sufficiently detailed and transparent to enable reproducing the results.

The paper introduces a heavily documented approach using five critical components to find and create reservoirs parallel to major rivers. It leads to the identification of millions of potential projects, one of which is selected to illustrate the calculations (L126-133 & Fig. 3). This case example, as well as different locations shown in Supplementary Fig. 3 & 4, are prepared for illustrative purposes solely. Although it allows a better understanding of each project's design, it cannot serve as a criterion for **validating** the model. One is left with the question about how well does the model perform? Does the searching algorithm accurately and efficiently select off-river sites in regions where previous studies have reported potential reservoir locations, such as France (Ref. 16, especially in mountainous regions) or Brazil (Ref. 27)?

The methodology developed by the authors uses five critical components (L59-61), among which topographical and hydrological data have significant influence on project technical suitability. The construction of an upper reservoir "wherever in the landscape" (L199) has been said previously (Ref. 27) to help turning new dams projects more viable where there is no suitable geology for the construction of large conventional reservoirs in main river channels. However, the methodology does not integrate a **geological component** that checks if the construction of the upper reservoir is within a stable geological formation.

Returning to the subject of estimated costs for pumped-storage schemes, Ref. 26 points out that a

meticulous cost-benefit analysis of energy storage systems "requires consistent, updated cost data and a holistic cost analysis framework." The engineering design of the pumped-storage projects used in this study relies solely on a Master's thesis (Ref. 28) that is not a peer-reviewed report. Pumped-storage costs are calculated using very detailed data from Ref. 29-30. In this reference, "the stated prices represent the price level in January 2010", nearly ten years ago, while costs for renewable power generation vary from year to year (IRENA 2019). Moreover, cost analysis of pumped-storage technologies cannot be easily generalized as they are site-specific, warranting further sensitivity analysis to consider uncertainties (e.g. Ref. 26) and to determine what most strongly influences the storage potential.

APPROPRIATE USE OF STATISTICS AND TREATMENT OF UNCERTAINTIES

Treatment of uncertainties is however a difficult task in such a study where more than twenty items are listed as assumptions or limitations of the model (Supplementary Table 7), all of which are source of possible over- or underestimation of the storage potential and of uncertainties propagation through the model. It raises unanswered questions regarding for instance the quantitative assessment of how highly sensitive is the model to the resolution, topography, distance to a river, river discharge simulation, water availability (L61-62) or maximum percentage of the annual river flow (L95-100).

CONCLUSIONS (DISCUSSION)

The authors conclude on the vast technical and economical potential for water and energy storage with pumped hydro energy storage operated in a seasonal cycle. They could examine in greater detail (see Supplementary Table 7, C17) energy and water complementarity depending on latitude, energy demand and water availability, or energy generation from other renewables sources.

SUGGESTED IMPROVEMENTS

L21 "Pumped-storage systems" are mostly referred to as pumped-storage hydropower (PSH) or pumped hydro(electric) energy storage (PHES).

L21-24 Consider adding that pumped hydropower storage plants constitute the vast majority of electricity storage worldwide.

L25 "for daily or weekly cycles" and up to months (Ref. 4, 8 or 9).

L67 See L50 " The only current study looking at the global potential for SPS".

L69 "improved" compared to which previous estimation?

L86-87 " check the water availability of the river for storage" Couldn't this step more efficiently come earlier?

L88 "The hydrological data were used to restrict the size of the storage reservoirs"
To restrict or to remove reservoirs if they don't meet the hydrological ("without having a considerable impact on the overall river flow" or if river flow is constant for instance) or economic feasibility (is there a minimum size below which the project is not economically feasible?) criteria?
Did it happen?

See also L264-265 or Supplementary Table 3 "If the storage capacity is much higher than the amount of water available, the estimated cost of storage tends to zero, as the reservoir will never

fill up." & L274 Is the project therefore cancelled?

L108 I do not understand why water and energy storage projects are considered separately. Isn't the principle of SPS projects to provide multiple and complementary services (L188-192)? On which criteria are they distinguished? Is the total number of potential storage projects the sum of the two?

L118 Why showing only energy storage projects cost distribution?

Supplementary Table 4 Why not providing all the values for both types of project (e.g. L9C8 average storage volume for energy storage projects, etc.)?

L111-112 & Supplementary Table 7 L4C2 The "land-value of 41,000 \$/ha" is indeed ten times higher than the one used in Ref. 10. Is this value realistic?

L134 & L136-137 & Fig. 4 Could you justify the water and energy storage thresholds of respectively 0.2 \$/m³ and 50 \$/MWh?

L148 Why are dam and land costs excluded? cf. L109 " Critical components of the SPS project costs are the dam and tunnel".

L150 Leaves us with some unanswered questions: how does it compare to other short term energy storage technologies?

L159-160 Fig. 4e, which is almost yellow everywhere, actually does not allow to conclude that the percentage varies "considerably" from case to case.

It would be of interest to know how many sites built "wherever in the landscape" (L199) are not intersecting a tributary river.

L159-161 Calculation of these percentages and consideration of their related costs are not described neither in the hydrological analysis stage nor in the estimate cost stage (Methods & Supplementary Table 3).

L162 Is it worth mentioning three over more than 1,000 projects?

L167-168 Which kind of "energy storage in Brazil"?

L186-187 Why 15 \$/MWh instead of 50 \$/MWh?

L188-190 Are these average values?

L200-201 A huge potential but with a unequal spatial distribution.

L207-208 Regional studies such as Ref. 16 already try to eliminate irrelevant sites due to conflicts with existing land use.

L225-226 Are the GRIN and PCR-GLOBWB river networks spatially compatible?

L227-230 I could not find further explanations on these procedures in Supplementary Table 2, which simply refers to the same two references.

L291 Is it always a significant benefit to reduce flow seasonality?

Supplementary Tables (L line, C column)

Supplementary Table 1

L6C2 How can energy be stored both in winter and summer?

L11 The implicit comparison to conventional reservoir dams tends to minimize seasonal pumped-storage environmental impacts or disadvantages.

L15C2 What about dams in cascade impacts on the main river flow (Ref. 31 or Fig.6 in Ref. 10)?

L16C2 Reference for this statement?

L17C2 Seasonal water and energy storage will not be always complementary/compatible to flooding control.

L18C2 Is this not a minor issue? Do you have examples of SPS plant channels used for waterways transport?

Supplementary Table 2

L2C4 Specify 56°S to 60°N

What is meant by "5x5 data"?

L4C4 Indicate the period of analysis for hydrological data (1960-2010). Which simulation run? With human activities?

L6C4 The methodology used in Ref. 28 is based on "existing hydropower reservoirs and existing power plant(s)".

Supplementary Table 3

L6C2 Why a 1,620 km² threshold?

Supplementary Table 7

L8C2 " Thus, there are sections of the river where the hydrology data give a much lower value." Much lower than what?

"In these cases, it assumes the average value of the particular river Strahler flow in a 5° resolution." Please clarify.

What about uncertainties associated with global hydrological modelling?

L9C2 Other aspects that are not included in this "rough" cost estimate: true social-environmental and cultural costs, lifetime costs, etc.

L16C2 Hydrological and environmental impacts on the tributary river?

L17C2 What about mid-latitude regions? One can think of other combinations that may decrease the SPS potential, such as a dry season with high solar power generation but low flows; a wet season with low solar power generation, high electricity demand and flooding risk. See also Supplementary Table 1 L6C2 "Countries in mid and high latitudes tend to have a seasonal electricity demand profile, consuming more electricity summer for cooling and during the winter for heating purposes, respectively."

L23C2 A greater caution would be required, as regards the potential negative impacts (e.g. wouldn't it prevent groundwater recharge?): examples of proven efficiency? References?

TECHNICAL EDITS

L14 " methodology **based on**"

L15-17, L106-107, L112, L127, L134, ..., Specify US\$ instead of \$.

L51 "might not **be**"

L66 " for **conventional** reservoir **dams** estimation"

L78 No comma after "framework".

L79 "Supplementary Tables 2, 3 and **4**" Table 7 should be numbered 4.

L82-83 "from **the grid cell**"?

L83-87 "checks" "removes" "finds" "estimates" etc.

L84 "the flooded side of the dam"

L88-90 Should go with the next paragraph, dedicated to hydrological data.

L100 Should be Supplementary Fig. 1, as first cited figure in the text.

L104 Supplementary Figs. 3 and 4 (Proposed SPS pipelines & dams) do not illustrate the point (L103-104 "The SPS world potential model identified more than 5.1 million potential projects, all of which have a fixed generation/pumping capacity of 1 GW ").

L115-116 151.7 meters is for energy storage projects.

L138 Rephrase "The water stored [...] also stores water".

L143 "the cost of energy storage with natural gas"

L144 "with cascade"

L144-145 (Fig. 4c).

L159 "Percentage of inflow" from the tributary river

L160 Rephrase "water that requires to be pumped".

L162 from which river?

L165 "varies"

L182-187 (Fig. 4).

L186 "energy storage costs" with cascade?

L188 "SPS projects"

L201 Add a reference.

L218 "56°S"

L224-225 Delete "This is used to give a better estimate of the tunnel length connecting the river and the reservoir." (see L240-241).

L226 "hydrological data taken from the PCR-GLOBWB global hydrological model"
"which are"

L235-236 Delete "at the same resolution with the intention of finding the location of the rivers in the topographic data".

L243, L244, L248, Supplementary Table 3 L3C2, Supplementary Table 4 L6C1, Supplementary Table 6 L1C4 "Strahler stream order" instead of "Strahler".

L247 "56°S"

L249 "consists"

L263 " which is a small portion of the river flow total river flow and results in"

L269 Most of previous paragraph (L260-265) was already part of the hydrological analysis stage.
Think of reorganising text.

L271-272 Repetition of previous sentence.

L275 "deregulates"

L277 "S_v is the seasonality index, I_v is the inter annual variation index"

Eq. 1 & 2 & L279 N_m is the number of months and N_Y, the number of years.

Delete L280-284.

L285 "services"

L286 "variations"

L295-296 Q_A and Q must have the same unit.

L299 "storage is"

L302 Add a comma after "\$".

L305 "E_Rwc and E_Rwoc"

FIGURES & TABLES

Fig. 1 Origin of the data plotted on this figure?

Zn-Br in the legend instead of VR.

L33 "seconds to year" (according to the figure).

Fig. 2 Font size is often too tiny.

Fig. 2b c frame "Strahler"

Fig. 2d What does "Dam Height = 50" mean? Unit?

Fig. 2g Add an arrow back in case of constant river flow from hydrological analysis?

Fig. 2h Units in legend?

Fig. 3 Font size is often too tiny.

Fig. 3a Turbine

Fig. 3b US\$

Energy storage costs with cascade.

Fig. 3c Change for geographic coordinates.

Reservoir is difficult to see but appears blue and not purple.

Fig. 3d Units in legend?

Fig. 4 Font size is too tiny.

Australia and Europe lines are the same colour (a b c d right side).

Fig. 4e "Percentage of the reservoir that is filled with the tributary river inflow" (caption).

y-axis legend "Tributary river flow percentage (%)" is therefore ambiguous.

Supplementary Fig. 2

L53 "extracts"

L55 "withdrawals"

Supplementary Figs. 3 and 4

Add legend & coordinates.

Supplementary Fig. 3 is not entirely clear.

Supplementary Tables (L line, C column)

Supplementary Table 1

L6C2 "consuming more electricity in the summer for cooling"

No link between island electricity generation in L10C1 and costs in L10C2.

Supplementary Table 2

C6 Reference numbers are inappropriate, please check and correct (+3).

Supplementary Table 3

Erroneous "Links in the paper": starting from L2, should be Fig. 2a / Fig. 2b / Fig. 2d / Fig. 2e / Fig. 2f / / Supplementary Tables 4 and 7 / ...

L3C2 "This stage looks for a river"

L6C2 No commas before and after "which side of the dam".

L7C2 "within the reservoir"?

L8C2 "This section calculates"

Supplementary Table 4

L14C1 S_v from Eq. 1.

L15C1 I_v from Eq. 2.

Same in Supplementary Table 5 L1C7 & L1C8.

L15C1 "variability" or "variation"

Supplementary Table 5

Please specify "in Tibet, China" in the table caption.

L1C4, L1C6, L1C8 Add "cost".

Supplementary Table 6

Please specify "in Tibet, China" in the table caption.

Supplementary Table 7

L8C2 " The hydrological data is in 0.1° resolution" (Supplementary Table 2, L4C3)

"much lower than the 15 sec topography resolution"

L15C1 "Open-loop SPS"

L15C2 "Closed-loop SPS require two large reservoirs"

"Additionally, there would be no hydropower plants in cascade that increase energy storage without additional costs."

L16C2 "from the tributary river"

L18C2 " The needs for energy and water storage are"

REFERENCES

The references section could have been checked more carefully for accuracy before submission and must be improved: there are duplicates, second-hand references, erroneously numbered references (Supplementary Tables 1 & 2) and not up-to-date references. Details below.

L21-24 Rehman et al. (2014) provide an extensive review of pumped-storage systems. Rehman

S., Al-Hadhrami L.M., Alam M.M. Pumped hydro energy storage system: A technological review. *Renewable and Sustainable Energy Reviews* **44**, 586–598 (2015).

L329-330 2. International Renewable Energy Agency (IRENA). Renewable power generation costs in 2014. (**2015**).

IRENA 2019 report was recently published (May 2019) and could be of interest for this study. IRENA 2018 report provides an estimation of global installed capacity for year 2017 but its source is the DOE Global Energy Storage Database (<http://www.energystorageexchange.org/>). See also IRENA renewable energy statistics (July 2019).

L331-332 This is a second-hand reference. Source is again the DOE (see previous comment).

L333 4. International Electrotechnical Commission. Electrical Energy Storage: White Paper.

L366-367 18. Stocks, M. *et al.* A global atlas of pumped hydro energy storage (2019). Available at: <https://nationalmap.gov.au/renewables/#share=s-oDPMo1jDBBtwBNhD>.

L368-369 19. Gimeno-Gutiérrez, M. & Lacal-Aránategui, R. Assessment of the European potential for pumped hydropower energy storage. *JRC Scientific and Policy Reports* (2013).

L370-371 20. Lacal-Aránategui, R., *et al.* & Leahy, N. Pumped-hydro energy storage: potential for transformation from single dams. *JRC Scientific and Technical Reports* (2012).

L368-369 19. and L370-371 20. could be replaced by Gimeno-Gutiérrez M., Lacal-Aránategui R. Assessment of the European potential for pumped hydropower energy storage based on two existing reservoirs. *Renewable Energy* **75**, 856-868 (2015).

L390 28. Rognlien, L. Pumped Storage Development in Øvre Otra, Norway. MSc Thesis, Norwegian University of Science and Technology (2012).

L391-392 29. and L393 30. are the same. Cost base for hydropower plants (With a generating capacity of more than 10,000 kW). Published by: Norwegian Water Resources and Energy Directorate. Editor: Jan Slappgård. Authors: *SWECO Norge AS*.

L399 33. *Inage, S-I*

L406-408 Please delete the reference to this website.

L409-410 37. International Energy Agency. Energy Technology Perspectives: *scenarios* & Strategies to 2050. (2008).

L414 Ref. 39. is redundant with the previous one (Ref. 38).

L421-422 Where is Ref. 42 cited?

Supplementary Table 1 (L line, C column):

L3C1 Ref. 24 not appropriate here.

L4C1 Ref. 31 not appropriate here (but in L3C1).

L5C1 Ref. 34 focuses on a case study for the United States, not on high latitudes

L6C2 Ref. 35 not appropriate here (but in L4C1).

L8C1 Ref. 36 is not related at all to low energy security; ref. 37 not appropriate here (but in L10C2).

L10C1 Ref. 40 & 41 (Canary Islands) instead of 38 & 39?

Reviewer #2 (Remarks to the Author):

The strength of the paper is the global analysis for adequate sites for seasonal pump storage projects. The logic is to find favorable terrain conditions for the construction of a dam to store water (upper reservoir) nearby a river with enough water flows for the seasonal operation. The water is pumped from the river into the upper reservoir "parallel" to the river system, at a higher elevation in periods of low demand or favorable hydrology. It is then used for producing electricity in periods of high demand or insufficient water flows in the river.

This approach has several alleged advantages with respect to the most common alternative of damming the river, such as the reduction of the socioenvironmental impact due to the smaller flooded areas, the fact that the river connectivity is unaffected and the possibility to supply fresh water to nearby communities.

The paper expands a previous work (reference 17) that focused on closed PHS systems that also investigated promising sites for PHS on a global scale. The main difference, however, is the configuration of the PHS system. Reference 17 investigates closed system, where both lower and upper reservoirs are created by dams (sometimes many dams are built to avoid points of "leakage" on the terrain) whereas the present paper focuses, as mentioned, siting one dam parallel to a river.

The structure of paper is not ideal. I spent a long time trying to understand the workflow of Fig.2. I was only able to decipher it could after reading section "Methods", which comes much later in the paper. It is also odd that "Discussion" precedes "Methods". Work needs to be done to reorder the sections for the sake of clarity.

Another important remark is that most assumptions, parameters and formulas are either not explained or insufficiently justified. References could be made when possible. To name a few examples, why is the maximum volume of the reservoir equal to 11% of the annual river flow? (it is evident that this is the cap from formula 3, but the rationale of this formula should be mentioned). Why is the maximum dam length 7.2 km? Formulas (4) that present the calculation of economic performance of the seasonal pumped-storage (SPS) systems deserve further explanation, especially variable WS.

A web-based GIS interface with the results of this analysis would greatly strengthen the findings of this paper. It could also allow for the download of the processed results, as in the case of reference 18. This should not be a difficult task, considering the several possible open GIS platforms available.

The work has not evaluated the impacts of the dams with respect to the surrounding area because the work is based on topography only. If a dam floods a city, for example, it will not be discarded, despite being infeasible in real life. It is not clear many projects that have been screened would "survive" filters that would include other layers of information. The author recognizes this needs to be incorporated in the analysis in a future work.

Overall, I believe there is merit in the work, but it needs a thorough review before accepted.

I have also made comments and suggestions in the body of paper (Word file) for consideration.

Reviewer name: Rafael Kelman

1
2
3
4
5
6
7
8

Response to Reviewers

Global resource potential of seasonal pumped-storage for energy and water storage

Reply to Reviewer #1 comments:

Reviewer #1 comments	Reply to Reviewer #1 comments
Key results	
This work introduces the first high-resolution assessment of seasonal pumped hydro energy storage potential at global scale. Pumped-storage plants are used to store in a single project large amounts of water and energy in off-stream reservoirs during periods of high river flows or excess energy in the grid and to generate electricity by releasing water at times of highest system power demand. Pumped hydro energy storage is a mature and available technology, which has existed for a long time but is scarcely used on a seasonal fashion. Seasonal pumped-storage allows long-term storage of seasonal surplus energy, adding operating flexibility to the system while providing water flow regulation service at the same time. The model developed in this work identifies more than 1,000 suitable locations worldwide using a systematic approach, based on high resolution topographical data and discharge data from a global hydrological model. It also predicts the associated costs for water, short and long-term energy storage services. This study shows that there is a considerable potential for seasonal pumped-storage to provide competitive storage, although this potential varies substantially from region to region, being restricted to mountainous regions.	Dear Reviewer #1, Thank you for the detailed and valuable comments and suggestions to the paper and we are grateful for the substantial time that you spent genuinely trying to improve the paper. Implementing the suggestions has now considerably improved the paper in quality and scope.
Originality and significance	
The results presented in this work are of strong interest to the international community, regardless of discipline because of their social implications. As pumped storage hydropower is one of the most promising technologies for balancing seasonal fluctuations of intermittent renewables, identification of technically feasible sites for new installations and improved information on their costs are crucial. This work represents an advance in assessing global capacity for energy storage, likely to influence power system planners and foster the deployment of pumped hydro energy storage systems.	Thanks for the kind words.
The world potential of hydropower, which is currently the dominant renewable energy source, has already been evaluated. Previous studies also exist	Thanks for the kind words.

that quantify pumped hydro storage capabilities at regional scales, such as in Europe. As an attempt to comprehensively assess the world potential of seasonal pumped-storage using an upgraded model synthesising previous approaches, the results presented in this work are original. Also novel is the associated cost analysis presented here, despite the difficulties to provide an accurate global estimate of the costs.	
The authors nonetheless briefly mention the global atlas of pumped hydro energy storage (Ref. 18). In their report, Stocks et al. from the Australian National University found a total of more than 600,000 potentially feasible pumped hydro energy storage sites worldwide with a massive storage potential of about 23 million GWh. The identified sites are mostly closed-loop, comprising an upper and lower reservoir pair, which is ranked according to an approximate cost model. This very recently released database cannot simply be swept aside in a single sentence (L50-52) but need further discussion. One can wonder, for instance, to what extent the close-loop sites identified in the global pumped hydro atlas are complementary to or coincide with the open-loops considered in this study (Supplementary Table 7, L15C2).	We have looked at the reference from the Australian National University in details and found that the projects are not SPHS projects, they are daily or weekly PHS projects. For example, if we divide the total storage capacity of 23 million Gigawatt-hours (GWh) by the number of projects 616,000 gives an average storage capacity of 0.037GWh/project. Comparing with smallest project in this paper of 0.93TWh it can be seen that they are not looking for seasonal projects. So this is an important distinction with our study to be clarified. We changed the text to the one below: “One recent study investigates the global potential for PHS and assumes the construction of two reservoirs in a closed loop for daily and weekly operation. They found a global potential of 23×10^6 GWh in more than 600,000 plants, but the project sizes appear not be practical or viable for seasonal storage, water storage and does not include cost analysis^{22,23} (Supplementary Table 7)”. We have not included this closed loop option because they are designed to store energy and we are looking at energy and water storage solutions in this paper. We added new line named “Closed-loop SPHS” to Supplementary Table7 to add more discussion of the results found in the study. “Closed-loop SPHS power plants are not considered in this paper. They require two large reservoirs and only store energy. They do not have a substantial impact on the nearby rivers because they are not built on a river and the water inflow from a river only intends to complement the water losses due to evaporation. The inclusion of closed-loop SPHS in this study would considerably increase the world potential of energy storage with SPHS plants. Even though this arrangement would allow projects to be built far from major rivers, the need for building two large dams and reservoirs close to each other reduces the number of locations with appropriate topography which would make such projects viable. Additionally, there would be no hydropower plants in cascade that would contribute to the increase in energy storage without additional costs”. We also added new line named “SPHS in series” to Supplementary Table7 to increase the discussion on different SPHS arrangements. “There is also the possibility of building two SPHS reservoirs in series, where one of the reservoirs is connected to the main river (intermediate reservoir) and the other reservoir is

	connected to the intermediate SPHS reservoir (upper reservoir). This arrangement is interesting because it increases the possibility of increasing the total head of the water. Another limitation that this arrangement resolves is the fact that SPHS projects are usually limited to a generation head of 1200 meters. Having two SPHS in series could increase the overall generation head of the plant to 2400 meters. One example of such arrangement can be seen in the Limberg SPHS plant in Austria. This possibility would considerably increase the potential for water and energy storage presented in this paper, particularly the energy storage potential, due to the increase in overall generation head”.
--	--

This also points to a need for a clearer definition of a seasonal pumped-storage project in the introduction (L34-37, see Ref. 10). In addition to be helpful for those unfamiliar with the technology, it would help clarify its specificities. In particular, it is worth mentioning from the beginning that:

- "the river can have a small reservoir"** (Supplementary Table 7, L15C2),
- the energy storage potential also includes the energy generated at the dams in cascade downstream the plant (L137-139 & Supplementary Table 7, L2),**
- upper "sites are not restricted to rivers" (L198) but usually coincide with a tributary river feeding directly the reservoir (Supplementary Table 3, L5C2).**

We rewrote the paragraph that presents SPHS plants and added more references:

“A SPHS plant consists of a high-head variation storage reservoir built in parallel to a major river. During periods of low energy demand or high water availability, water is pumped into the reservoir. Stored water is released from the reservoir generating electricity when additional electricity generation capacity is required, or water is scarce. They can be compared to conventional reservoir dams, due to the possibility of regulating the river flow and increasing the hydropower generation on the hydropower dams in cascade¹⁵. SPHS plants have lower land requirements than conventional hydropower dams, for a comparable energy and water storage potential, because the off-river reservoir design permits higher hydraulic heads variations¹³. SPHS can also be attractive to deal with the load problems emerging from electricity consumption and supply seasonal variations and increasing use of intermittent sources of generation. The storage of water can also help to overcome water shortage problems. Because storage is also not near the main river, possible negative impacts of hydropower can be better managed (further details in Supplementary Table 1)”.

Clarity and context

The abstract is clear and accessible. It focuses attention on seasonal pumped-storage costs but could also mention the total number of selected projects with the lowest costs (L103-108) and point out the unequal spatial distribution of the considerable evaluated capacity (L17-18, L181-182). The authors should also state in the abstract that they present a rough (Supplementary Table 7, L9C2) global assessment of seasonal pumped-storage theoretical potential, "focusing on its technical potential" (L204-207) and without human considerations and climate change.	We changed the abstract according to your suggestions: "The risk of seasonal mismatches between electricity supply and demand is increasing due to expanded use of wind, solar and hydropower resources, which in turn raises the interest on low-cost seasonal energy storage options. Seasonal Pumped Hydropower Storage (SPHS) can provide long-term energy storage at a relatively low-cost and co-benefits in the form of freshwater storage capacity. Here, we present the first estimate of the global assessment of SPHS potential, using a novel plant-siting methodology based on high-resolution topographical and hydrological data. This estimate focuses on its technical potential and does not include human considerations and climate change. Our results show that SPHS costs vary from 0.007 to 0.2 US\$/m³ of water stored (over the 1,457 selected water projects), 1.8 to 50 US\$/MWh of energy stored and 370 to 600 US\$/kW of installed power generation capacity (over the 1,092 selected energy projects). This potential is unequally distributed with mountainous region having a considerable advantage. The estimated world energy storage capacity below a cost of 50 US\$/MWh is 17.3 PWh, approximately 79% of the world electricity consumption in 2017".
Indeed viability of such projects remains a major challenge, which somewhat limits the scope of this study. Despite the benefits of deploying seasonal pumped-storage (Supplementary Table 1), any new infrastructure will face conflicts with existing land use and significant costs might occur from mitigation measures. One might therefore ask to what extent these "restrictions that impact socio-economic feasibility" would influence the estimated costs and energy storage capacity, all the more since a single world capacity value is provided.	We added this concern to a line named "Restriction zones" in Supplementary Table 7. "The model assumes only technical aspects of SPHS projects. Restrictions such as population resettlement, social opinion, historical buildings and locations, environmentally protected areas, conflict zones were not included to the analysis. These restrictions would have a considerable impact on the estimated capacity and costs for water and energy storage. Further studies should be implemented and include considerations for the restriction mentioned above and other, which vary from region to region".
Validity, data & methodology	
The description of the first model stages (1 to 7) in the main body of text, in Fig. 2, in the Methods section and in the Supplementary Tables 2 and 3 is quite redundant, without a lot of added value. The text should be more concise regarding these stages but more explicit in reporting cost estimation methodology, which is not sufficiently detailed and transparent to enable reproducing the results.	We agree with this concern and considerably increased the description of the assumptions of the model in Supplementary Tables 2, 3 and 7.
The paper introduces a heavily documented approach using five critical components to find and create reservoirs parallel to major rivers. It leads to the identification of millions of potential projects, one of which is selected to illustrate the calculations (L126-133 & Fig. 3). This case example, as well as different locations shown in Supplementary Fig. 3 & 4, are prepared for illustrative purposes solely. Although it allows a better understanding of each project's design, it cannot serve as a criterion for validating the model. One is left with the question about how well does the model perform? Does the searching	We added references that compare the methodology implemented with real examples of reservoirs. We added a new line named "Model Validation" in Supplementary Table 7. "We validated different aspects of the model separately. For example 17,18 also used SRTM topographic data to estimate the volume of its reservoirs and validates their reservoir storage capacity with a sample of reservoirs or lakes with known volumes, and shows that the error in estimating the water storage is small. The reference from where the costs analysis was taken from ¹² is validated by the extensive experience of construction of hydropower plants in Norway.

algorithm accurately and efficiently select off-river sites in regions where previous studies have reported potential reservoir locations, such as France (Ref. 16, especially in mountainous regions) or Brazil (Ref. 27)?	The final costs for energy storage with PHS are similar to costs presented in the literature of 400 to 800 €/kW and 5 to 150 €/kWh¹⁹.
--	--

The methodology developed by the authors uses five critical components (L59-61), among which topographical and hydrological data have significant influence on project technical suitability. The construction of an upper reservoir "wherever in the landscape" (L199) has been said previously (Ref. 27) to help turning new dams projects more viable where there is no suitable geology for the construction of large conventional reservoirs in main river channels. However, the methodology does not integrate a geological component that checks if the construction of the upper reservoir is within a stable geological formation.	We added a line named "Geological formation" to the "Uncertainties" section of Supplementary Table 7. "This work does not include the geological composition and structure of the topography. An appropriate geological formation is crucial for the construction of SPHS reservoirs. This is because SPHS reservoir suffers from great pressure due to the high column of water that could be as high as 250 meters in the model. Another issue is that this stress varies throughout the year with fluctuation in level of the reservoir, which could result in fatigues in the geological formation. Thus, the locations considered for the construction of SPHS reservoirs should not have geological fractures and should have a composition that can withstand these pressures and fatigues".
---	--

Returning to the subject of estimated costs for pumped-storage schemes, Ref. 26 points out that a meticulous cost-benefit analysis of energy storage systems "requires consistent, updated cost data and a holistic cost analysis framework." The engineering design of the pumped-storage projects used in this study relies solely on a Master's thesis (Ref. 28) that is not a peer-reviewed report. Pumped-storage costs are calculated using very detailed data from Ref. 29-30. In this reference, "the stated prices represent the price level in January 2010", nearly ten years ago, while costs for renewable power generation vary from year to year (IRENA 2019). Moreover, cost analysis of pumped-storage technologies cannot be easily generalized as they are site-specific, warranting further sensitivity analysis to consider uncertainties (e.g. Ref. 26) and to determine what most strongly influences the storage potential.	Whilst we have used the same cost parameters for all plants across the world, the wide range of inputs and complexity of equations takes this beyond what we would consider a generalized analysis. Every plant has a unique topography, hydrology and sizing, resulting from the equations to estimate the cost mentioned below. This is far beyond many technological assessments that simply use fixed \$/kW capacity values. Construction costs (e.g. labour, materials) will vary from place to place, but in using this method we now have a globally comprehensive and comparable sample that gives an overview of the global techno-economic potential. We substantially expanded the lines "Pumped Storage Costs" and "Tunnelling Design" in the Supplementary Table 2. "Pumped Storage Costs" "This reference gives very detailed data on pumped-storage costs, such as dam, tunnels, excavation, electrical equipment and turbine costs. The model assumes most cost estimates proposed by the reference 15. It also assumes only one type of construction design for each of the components of the SPHP plant. This is because, it would be complex to create a model that compares different designs for each component to find the most optimum one. This given a good preliminary estimate of the final costs. For the construction of the dam, the model assumes a rockfill dam with central moraine sealing, as described in Fig. B.1.115. For the construction of the tunnels it assumes drill and blast, as described in Fig. B.1.415. The penstock costs include the costs of digging the tunnel (Fig. B.9.215) and the cost of the embedded steel pipes (Fig. M.6.C15). The excavation varies with the generation head and the installed capacity, as described in Fig. B.10.115. The turbine assumed is Francis, as described in Fig. M.1.b15 and Fig. M.4.A15. The selection of the turbine, also depends on the generator, as described in Fig.E.8.2.a15. For the optimization of the turbine/generator system, the
--	---

	costs of different rotation speeds, as described in Fig. E.1.1a15 and Fig. E.8.1.b15, are compared to the average generation head and flowrates under analysis and the cheapest option is selected. Note that one turbine/generator system is proposed per tunnel”. “Tunnelling Design” “projectsThe methodology used to optimize the construction of the tunnels was taken from 16. This methodology consists of comparing the capital costs of construction of the tunnels, such as the diameter and number of tunnels, and the costs of operating the plants. The cost of operating the plants depends considerably on the energy losses due to friction in the tunnels. The bigger the diameter and number of tunnels the more efficient is the plant”.
--	--

Appropriate use of statistics and treatment of uncertainties	
---	--

Treatment of uncertainties is however a difficult task in such a study where more than twenty items are listed as assumptions or limitations of the model (Supplementary Table 7), all of which are source of possible over- or underestimation of the storage potential and of uncertainties propagation through the model. It raises unanswered questions regarding for instance the quantitative assessment of how highly sensitive is the model to the resolution, topography, distance to a river, river discharge simulation, water availability (L61-62) or maximum percentage of the annual river flow (L95-100).	We added a line named “Resolution” to the “Uncertainties” section of Supplementary Table 7. “The resolution of the topography was reduced from 3 sec to 15 sec to increase the computational speed of the model. With a resolution of 15 sec, the model took one month to finish calculations. With a resolution of 3 sec, authors estimate that it would take more than two years to estimate the world potential for SPHS. A resolution of 15 sec, is equivalent to 450 meters in the equator, this low resolution impacts mainly the estimation of the SPHS dams. 450 meters is similar to the length of a medium sized dam. This low resolution fails to consider the complete profile of medium sized dams and frequently reduces the estimate of the height and length of the dam required to create the reservoir. This underestimates the costs of the dams for SPHS projects, particularly for dams with lengths shorter than 450 meters. This underestimation, however, reduces with the increase in dam lengths. Aside from dam size, costs are sensitive to the tunnel length. But overall, tunnel lengths are much longer than the topographical resolution so the error is small compared to total tunnel cost. Further explanation on the impact of the topographical resolution on PHS projects can be seen in ^{17,18}. The hydrological resolution of 6 mins is acceptable because the flow variations per pixel within a river is small fraction of the total river flow”.
---	--

Conclusion (Discussion)	
--

The authors conclude on the vast technical and economical potential for water and energy storage with pumped hydro energy storage operated in a seasonal cycle. They could examine in greater detail (see Supplementary Table 7, C17) energy and water complementarity depending on latitude, energy demand and water availability, or energy generation from other renewables sources.	We added the paragraph below to the discussion section: “The needs for energy and water storage with SPHS plants should be complementary. This is because during the dry season there will be low volumes of water available to be used for energy storage. This complementarity is usually the case in high latitude countries, where during the summer river flow is higher due to ice melting and energy demand is lower compared to the winter. Inter-tropical regions with abundant hydropower generation also have complementarity, where during the wet season there is high water availability and hydropower generation. However, there are regions and countries where the need for energy and water storage is not complementary, for example, in the inter-tropical regions
---	--

	without hydropower generation, where the summer and wet season is the period with highest electricity demand due to air conditioning. In cases where energy and water storage need are not complementary, SPHS should not be considered as an energy and water storage alternative”.
Suggested Improvements	
L21 "Pumped-storage systems" are mostly referred to as pumped-storage hydropower (PSH) or pumped hydro(electric) energy storage (PHES).	Done, we changed to pumped hydropower storage (PHS) as you proposed in the comment below.
L21-24 Consider adding that pumped hydropower storage plants constitute the vast majority of electricity storage worldwide.	Done, thanks.
L25 "for daily or weekly cycles" and up to months (Ref. 4, 8 or 9).	Done, thanks.
L67 See L50 " The only current study looking at the global potential for SPS".	Well observed, we are also looking into the global potential for SPS. We changed to “A recent study looks at the global potential for SPHS and assumes the construction of two seasonal reservoirs in close loop, which might not practical or viable, and does not include cost analysis ^{17,18”.}
L69 "improved" compared to which previous estimation?	There is no previous analysis. We removed “improved”.
L86-87 " check the water availability of the river for storage" Couldn't this step more efficiently come earlier?	I understand your concern. We have the hydrological data before all the modeling steps, so why add a Hydrological Analysis now? This stage checks if the there is enough water to fill up the proposed storage reservoir. In other words it compared the available storage potential with the water available to store. For example, If the storage reservoir is very large and the water available is small, then the cost of storage of the large reservoir will be high, thus a reservoir with a smaller dam and reservoir will make more financial sense. This step could come before “Estimate Project Cost”, but it should be after “Reservoir Storage Capacity”. To avoid the confusion we changed the name of the step to “Reservoir, Hydrology Comparison”. We rewrote the line in Supplementaty Table 3: “The hydrology is included in the analysis to limit the water and energy storage capacity of the SPHS projects according to the availability of water in the main river. The maximum water storage capacity is limited to 11% of the river flow. If the storage capacity is much higher than the amount of water available, the estimated cost of storage tends to zero, as the reservoir will never fill up. In other words, this section does not remove the project that does not have enough water to fill up the reservoir. It calculates the cost of energy and water storage with a large reservoir, even if the water available in not enough to fill the reservoir. For example, if the reservoir is two times larger than the water available, then the cost of energy storage will be higher than if there was enough water to fill the reservoir. Thus, the reservoir becomes too expensive and is not selected. The same reservoir with a smaller dam is selected instead, as the cost of the dam and flooded area are smaller. In other words, the project is not cancelled it is just not selected”.

L88 "The hydrological data were used to restrict the size of the storage reservoirs"
To restrict or to remove reservoirs if they don't meet the hydrological ("without having a considerable impact on the overall river flow" or if river flow is constant for instance) or economic feasibility (is there a minimum size below which the project is not economically feasible?) criteria? Did it happen?
See also L264-265 or Supplementary Table 3 "If the storage capacity is much higher than the amount of water available, the estimated cost of storage tends to zero, as the reservoir will never fill up." & L274 Is the project therefore cancelled?

This section does not restricts the project that does not have enough water to fill up the reservoir. It calculates the cost of energy and water storage with a large reservoir, even if the water available in not enough to fill the reservoir. For example, if the reservoir is two times larger than the water available, then the cost of energy storage will be around two times higher than if there was enough water to fill the reservoir. Thus, the reservoir because too expensive and is not selected. The same reservoir with a smaller dam is selected instead, as the cost of the dam and flooded area is smaller. In other words, the project is not cancelled it is just not selected. We added to Supplementary Table 3:
"In other words, this section does not restricts the project that does not have enough water to fill up the reservoir. It calculates the cost of energy and water storage with a large reservoir, even if the water available in not enough to fill the reservoir. For example, if the reservoir is two times larger than the water available, then the cost of energy storage will be around two times higher than if there was enough water to fill the reservoir. Thus, the reservoir because too expensive and is not selected. The same reservoir with a smaller dam is selected instead, as the cost of the dam and flooded area is smaller. In other words, the project is not cancelled it is just not selected".

I do not understand why water and energy storage projects are considered separately. Isn't the principle of SPS projects to provide multiple and complementary services (L188-192)?

There are 5.1 million water and energy projects considered. From these projects their water and energy costs are estimated. It turns out that for each 1 arc degree resolution there are 1,457 projects with water storage costs lower than 0.2 \$/m3 and 1,092 energy projects with energy storage cost lower than 50 \$/MWh. I added this to the main paper.
"This consists of 1,457 water storage projects with water storage costs lower than 0.2 \$/m3 and 1,092 energy storage projects with energy storage cost lower than 50 \$/MWh.".
 Also added a new line to the Supplementary Table 7 "Water and energy projects:
There are 5.1 million water and energy projects considered. From these projects, their water and energy storage costs are estimated. It turns out that for each 1 arc degree resolution there are 1,457 projects with water storage costs lower than 0.2 \$/m3 and 1,092 energy projects with energy storage cost lower than 50 \$/MWh".

On which criteria are they distinguished?

Each project can be used to provide multiple and complementary services, however, it would be confusing to combine both services in the paper as each location was different water and energy storage demands. Thus, we decided to present the storage costs individually. If the SPHS plant is used for energy and water store, then the cost for energy and water storage sill reduce as both services will contribute to the viability of the project. We added a new line to the Supplementary Table 7 "*Multiple Complementary Services: Each project can be used to provide multiple and complementary services, however, it would be confusing to combine both services in the paper as each location was different water and energy storage needs. Thus, we decided to*

	present the water and energy storage costs individually. If the SPHS plant is used for energy and water storage, the cost for energy and water storage will reduce as both services will contribute to the viability of the project”.
Is the total number of potential storage projects the sum of the two?	No, some of the energy projects are the same as the water projects. This mainly depends on the head of the plant. Cheap energy storage projects have high heads. Water storage projects are not affected so much by the head variation between the lower and the upper reservoir. Added to the main text “(some of the water projects consists of the same energy projects)” and added a new line to the Supplementary Table 7 “Water and energy projects Some water and energy projects presented in Fig. 4 consist of the same project. However, most of them are different because cheap energy storage projects usually have high heads. The higher the head, the more energy the SPHS plant stores. Water storage projects, on the other hand, are not affected so much by the head variation between the lower and the upper reservoir”.
L118 Why showing only energy storage projects cost distribution?	Thanks for noticing, Fig. 3a refers to both water and energy storage costs. We corrected the main text: “Fig. 3 SPHS costs and description. a, Water and energy SPHS project cost distribution shows that the most expensive components tend to be the tunnel and dam.”.
Supplementary Table 4 Why not providing all the values for both types of project (e.g. L9C8 average storage volume for energy storage projects, etc.)?	We agree with you. It is interesting for comparing how a project designed to store water would perform storing energy, and how projects designed to store energy performs storing water. We added this information to the table.
L111-112 & Supplementary Table 7 L4C2 The "land-value of 41,000 \$/ha" is indeed ten times higher than the one used in Ref. 10. Is this value realistic?	The authors decided to use high land cost because one of the most relevant draw backs from the construction of SPHS projects is the need to create a large reservoir. Setting a large land cost forces the model to look for projects which have small flooded areas, and thus lower social and environmental impacts. We also decided to choose a fixed land cost because different countries have different policies involving the construction of reservoirs, which proved to be difficult to quantify in the model. A fixed and high land cost assumes that the appropriate social and environmental measures can be paid for the construction of the SPHS reservoir. We added this text to the Supplementary Table 7. “One of the most relevant drawbacks for the construction of SPHS projects is the need to create a large reservoir. Setting a high land cost forces the model to look for projects with small flooded areas and, thus, lower social and environmental impacts. A fixed land cost was used because different countries have different policies involving the construction of reservoirs, which proved to be difficult to quantify in the model. A fixed and high land cost assumes that the costs for appropriate social and environmental measures in the construction of the SPHS reservoir is included in the project cost”.
L134 & L136-137 & Fig. 4 Could you justify the water	We added a line in Supplementary Table 7:

and energy storage thresholds of respectively 0.2 \$/m³ and 50 \$/MWh?	“Water and energy storage costs thresholds The water and energy storage costs thresholds were set with the intention of reducing the total number of projects presented in Fig. 4 to around a thousand, for a cleaner presentation of the results and to focus on possibly viable projects. The threshold cost for water storage of 0.2 \$/m³ is used based on an order of magnitude comparison with the cost of seawater desalination of 1 \$/m³. However, a SPHS plant still requires electricity to store the water, which is not included in the 0.2 \$/m³ cost. The threshold cost for energy storage of 50 \$/MWh is used based on an order of magnitude comparison with gas based thermoelectric generation, which varies from 10-50\$/MWh. However, note that a gas power plants generates electricity, while a SPHS plant only stores energy”.
L148 Why are dam and land costs excluded? cf. L109 " Critical components of the SPS project costs are the dam and tunnel".	We added a line in Supplementary Table 7: “This paper assumes that all projects proposed have 1 GW of installed capacity. The cost for the installed capacity of the SPHS plant in Fig 4d only includes the costs of the tunnels, turbine, excavation and electric equipment, excluding the dam and land cost. This is purposely set to allow the reader to design their own SPHS projects. For example, if the reader wants to propose a project with 2 GW, they will use the cost for energy storage project in Fig 4c and add the cost in Fig 4d to the project. Alternatively, if the user want to develop a project with 500 MW, he or she will deduct half the cost presented in Fig 4d in the project”.
L150 Leaves us with some unanswered questions: how does it compare to other short term energy storage technologies?	This is a very interesting question, however, the word limitation in the paper does not allow for a comprehensive comparison so we added a line in Supplementary Table 7 to expand on this issue: “Comparison with other short term energy storage technologies: Currently, PHS is the most economical alternative for short term energy storage, however, with the reduction in the cost of batteries it is possible that batteries will become cheaper the PHS plants. SPHS can provide both long and short term storage services. This combination of services could further increase the viability of SPHS for short term storage, due to the provision of two services with one plant. On the other hand, SPHS plants with short heads or long tunnels, i.e. with high installed capacity costs per GW, might not be able to compete with the cost of short energy storage provided by the batteries. In these cases, the operation of SPHS plants would focus on storing energy only for the long term and in crucial moments of the grid when peak generation is required”.
L159-160 Fig. 4e, which is almost yellow everywhere, actually does not allow to conclude that the percentage varies "considerably" from case to case.	You are right, we removed “considerably”.
It would be of interest to know how many sites built "wherever in the landscape" (L199) are not intersecting a tributary river.	We agree that the statement is confusing, all reservoirs would be built in a tributary river, small or large. So we decided to remove the text “because sites are not restricted to rivers but can be built wherever in the landscape”.
L159-161 Calculation of these percentages and	You are right. This is one of the most controversial issues in

consideration of their related costs are not described neither in the hydrological analysis stage nor in the estimate cost stage (Methods & Supplementary Table 3).	the paper. We decided not to include the hydropower potential in the model because it is complicated to compare hydropower and energy storage. Hydropower generates electricity and SPHS stores energy. Including both alternatives in the model would confuse the analysis. We expanded the Hydropower potential column in Supplementary Table 7: “Even though the tributary river flow is able to fill up the reservoir without the need for pumping water from the main river in some SPHS plants, as shown in Fig 4e, the model does not add the additional hydropower generation from the tributary rivers where the upper reservoir of the SPS is built. This is because it is complicated to compare hydropower and energy storage. Hydropower generates electricity and SPHS stores energy. Including both alternatives in the model would confuse the analysis”.
L162 Is it worth mentioning three over more than 1,000 projects?	Added, thanks.
L167-168 Which kind of "energy storage in Brazil"?	Added “hydropower”, thanks.
L186-187 Why 15 \$/MWh instead of 50 \$/MWh?	It was a silly mistake. Thanks a lot for spotting this.
L188-190 Are these average values?	These values are close to average values.
L200-201 A huge potential but with an unequal spatial distribution.	Interesting observation, added to the paper.
L207-208 Regional studies such as Ref. 16 already try to eliminate irrelevant sites due to conflicts with existing land use.	We removed “The addition of these restrictions is proposed for future work and regional studies” and replaced by “Regional studies such as ¹⁶ already try to eliminate irrelevant sites due to conflicts with existing land use”.
L225-226 Are the GRIN and PCR-GLOBWB river networks spatially compatible?	They are not, we had developed a methodology to match the high resolution data (PCR-GLOBWB) with the low resolution data (GRIN). We describe the methodology in Supplementary Table 2. “As the GRIN and PCR-GLOBWB data have different resolution, a methodology was created to increase the resolution of the PCR-GLOBWB data. This methodology consists of giving a single hydrological flow for each river Strahler stream order higher than 7 in each 5 degrees section. This is performed by finding the highest river Strahler stream order of each PCR-GLOBWB 6 min resolution, then taking an average of the hydrological flows for each river Strahler stream order number. A drawback of this methodology is that the river flow for each Strahler stream order in a 5 degree section will be constant. However, errors involving the topographic difference between the data are minimized. In order to improve the results using this methodology, it could have been applied to smaller sections of 1 degree or less. Assuming that there are uncertainties associated with the PCR-GLOBWB global hydrological model, the error of this methodology is small. Only rivers with a Strahler stream order above 7 were considered, as they have enough flow to justify the construction of a SPHP plant”.
L227-230 I could not find further explanations on these procedures in Supplementary Table 2, which	We gave more details on the design specifications taken from the reference in Supplementary Table 2.

simply refers to the same two references.	“, such as dam, tunnels, excavation, electrical equipment and turbine costs”. “, particularly paying attention to the optimization of the diameter of the tunnels considering the capital costs of the tunnel and the efficiency of the system”.
L291 Is it always a significant benefit to reduce flow seasonality?	No it is not. We had to create a methodology to justify a global need for storage, if not it would be difficult to propose reasonable projects for the globe. We added to Supplementary Table 7. “In order to create the model, a methodology to estimate how much water could be extracted from the river had to be created in a global scale. Given that the paper intends to resolve seasonal energy and water variation needs, the reduction in river seasonality was selected as a measurement of the amount of water that could be extracted from the river. This might not be the preferred option in the management of the water resources of a basin. For example, in Central Asia it is desired that the river flow continues with the least disturbance from hydroelectric dams because the demand for water in the countries downstream happens during the summer for irrigated agriculture. During the winter the demand for water is low. Other reasons why a reduced flow seasonality are due to environmental restrictions, river mouth and coastal sedimentation and other aspects”.
Supplementary Tables (L line, C column)	
Supplementary Table 1	
L6C2 How can energy be stored both in winter and summer?	The statement is confusing. We changed to: “Countries in high latitudes have a strong seasonal solar power generation profile. Seasonal storage allows energy to be stored in the summer and generate electricity during the winter, when there is lower solar generation”.
L11 The implicit comparison to conventional reservoir dams tends to minimize seasonal pumped-storage environmental impacts or disadvantages.	Added to the table in L15C2.
L15C2 What about dams in cascade impacts on the main river flow (Ref. 31 or Fig.6 in Ref. 10)?	Added to the table.
L16C2 Reference for this statement?	We have no reference to this statement, however, we strengthened the argument. “Storing the water parallel to the river, allows for a better control of the water quality in the reservoir, as it would not be directly affected by the fluctuations in water quality in the main river. Usually the water quality of a river deteriorates when the river flow is low and there is not enough water to dilute the pollutants in the water. In these low water availability periods the SPHS plant will not pumping water from the river. Water will be pumped into the river when the river flowrate is higher and the pollutants in the water are more dissolved. This will contribute to maintaining a better water quality in the reservoir. If the SPHS is still required to provide short term energy storage another small reservoir can be built close to the river, however, water from the lower reservoirs would not be exchanged with the river water to maintain the water quality in the SPHS reservoir”.

L17C2 Seasonal water and energy storage will not be always complementary/compatible to flooding control.	This is true, I expanded the approach to combine hydropower and SPHS for flood protection. “This combination consists of allowing the reservoirs in the main river to operate with low storage levels as the long term storage is performed by the SPHS reservoir parallel to the river. When the flooding inflow reaches the dam in the river it will be nearly empty and can store large parts of the flood waters. This combination guarantees that the system will store water and energy seasonally and that the dam in the river will have available storage volume to contain the flood”.
L18C2 Is this not a minor issue? Do you have examples of SPS plant channels used for waterways transport?	True, removed the statement. “SPHS plant channels could be also used for transport in waterways, combining the transport of water and goods. Additionally”.
Supplementary Table 2	
L2C4 Specify 56°S to 60°N	Added to the table.
What is meant by "5x5 data"?	We changed to “The reduction in resolution assumes the central point of 15 sec of the 3 sec data”.
L4C4 Indicate the period of analysis for hydrological data (1960-2010). Which simulation run? With human activities?	We added “This data combines estimated water availability and use over the period 1960–2010 and includes human activity. More details on the simulation can be in the reference”.
L6C4 The methodology used in Ref. 28 is based on "existing hydropower reservoirs and existing power plant(s)".	This is correct. This reference is used to use for the optimization of the diameter of the tunnels considering the capital costs of the tunnel and the efficiency of the system. It is convenient that they are compared with existing plant as the results from the analysis can be verified. We added to the table: “, particularly paying attention to the optimization of the diameter of the tunnels considering the capital costs of the tunnel and the efficiency of the system”.
Supplementary Table 3	
L6C2 Why a 1,620 km² threshold?	We wanted the model to have the least possible restrictions. The creation of the reservoir in the model is the step that takes the most computation time in the model. Given that the model was taking too long to complete we decided to reduce the allowed size of the reservoir. 1,620 seemed to be a large reservoir size that would not take too long to model the world potential. It turns out that the model took one month to finish calculations. If we would run the model again, we would lower the size of the reservoir.
Supplementary Table 7	
L8C2 " Thus, there are sections of the river where the hydrology data give a much lower value." Much lower than what?	We further explained the methodology used to reduce the resolution of the hydrological data. “There might be some uncertainties regarding the hydrological data. As the GRIN and PCR-GLOBWB data have different resolution, a methodology was created to increase the resolution of the PCR-GLOBWB data. This methodology consists of giving a single hydrological flow for each river Strahler stream order higher than 7 in each 5 degrees section. This is performed by finding the highest river Strahler stream order of each PCR-GLOBWB 6 min

	resolution, then taking an average of the hydrological flows for each river Strahler stream order number. A drawback of this methodology is that the river flow for each Strahler stream order in a 5 degree section will be constant. However, errors involving the topographic difference between the data are minimized. In order to improve the results using this methodology, it could have been applied to smaller sections of 1 degree or less. Assuming that there are uncertainties associated with the PCR-GLOBWB global hydrological model, the error of this methodology is small. Only rivers with a Strahler stream order above 7 were considered, as they have enough flow to justify the construction of a SPHP plant.”.
"In these cases, it assumes the average value of the particular river Strahler flow in a 5° resolution." Please clarify.	Changed the text.
What about uncertainties associated with global hydrological modelling?	Added.
L9C2 Other aspects that are not included in this "rough" cost estimate: true social-environmental and cultural costs, lifetime costs, etc.	Added to the table.
L16C2 Hydrological and environmental impacts on the tributary river?	Changed, thanks.
L17C2 What about mid-latitude regions? One can think of other combinations that may decrease the SPS potential, such as a dry season with high solar power generation but low flows; a wet season with low solar power generation, high electricity demand and flooding risk. See also Supplementary Table 1	Thanks, we added to the table: “Other cases of lack of complementarity can happen in locations where the dry season has higher solar power generation, or wet season with low solar power generation, or high electricity demand and flooding risk”.
L6C2 "Countries in mid and high latitudes tend to have a seasonal electricity demand profile, consuming more electricity summer for cooling and during the winter for heating purposes, respectively."	We changed the text in Supplementary Table 1 to avoid confusion: “Countries in mid and high latitudes tend to have a seasonal electricity demand profile. For example, they can consume more electricity during the summer for cooling or during the winter for heating purposes”.
L23C2 A greater caution would be required, as regards the potential negative impacts (e.g. wouldn't it prevent groundwater recharge?): examples of proven efficiency? References?	Added to Supplementary Table 1: “Ground water recharge The increased regulation of the river flow contributes to a continue supply of water. This could improve groundwater recharge as irrigation channels utilization would increases”.
Technical edits	
L14 " methodology based on"	Changed.
L15-17, L106-107, L112, L127, L134, ..., Specify US\$ instead of \$.	Changed.
L51 "might not be"	Changed.
L66 " for conventional reservoir dams estimation"	Changed.
L78 No comma after "framework".	Changed.
L79 "Supplementary Tables 2, 3 and 4" Table 7 should be numbered 4.	Changed.

L82-83 "from the grid cell"?	Changed.
L83-87 "checks" "removes" "finds" "estimates" etc.	Changed.
L84 "the flooded side of the dam"	Changed.
L88-90 Should go with the next paragraph, dedicated to hydrological data.	Changed.
L100 Should be Supplementary Fig. 1, as first cited figure in the text.	7.7% is the average of the values in SF 2.
L104 Supplementary Figs. 3 and 4 (Proposed SPS pipelines & dams) do not illustrate the point (L103-104 "The SPS world potential model identified more than 5.1 million potential projects, all of which have a fixed generation/pumping capacity of 1 GW").	Removed.
L115-116 151.7 meters is for energy storage projects.	Changed.
L138 Rephrase "The water stored [...] also stores water".	Fixed.
L143 "the cost of energy storage with natural gas"	Added.
L144 "with cascade"	Added.
L144-145 (Fig. 4c).	Added.
L159 "Percentage of inflow" from the tributary river	Added.
L160 Rephrase "water that requires to be pumped".	Rephrased.
L162 from which river?	Tributary.
L165 "varies"	Changed.
L182-187 (Fig. 4).	Added.
L186 "energy storage costs" with cascade?	Added.
L188 "SPS projects"	Added.
L201 Add a reference.	Added.
L218 "56°S"	Changed.
L224-225 Delete "This is used to give a better estimate of the tunnel length connecting the river and the reservoir." (see L240-241).	Deleted.
L226 "hydrological data taken from the PCR-GLOBWB global hydrological model"	Added.
"which are"	Changed.
L235-236 Delete "at the same resolution with the intention of finding the location of the rivers in the topographic data".	Deleted.
L243, L244, L248, Supplementary Table 3 L3C2, Supplementary Table 4 L6C1, Supplementary Table 6 L1C4 "Strahler stream order" instead of "Strahler".	Changed.
L247 "56°S"	Changed.
L249 "consists"	Changed.
L263 " which is a small portion of the river flow total river flow and results in"	Deleted.
L269 Most of previous paragraph (L260-265) was already part of the hydrological analysis stage. Think of reorganising text.	We changed the title of the section to " Storage dimensioning and costs ".
L271-272 Repetition of previous sentence.	Deleted.
L275 "deregulates"	Changed.
L277 "S _v is the seasonality index, I _v is the inter	Added.

annual variation index"	
Eq. 1 & 2 & L279 N_m is the number of months and N_Y, the number of years.	Changed.
Delete L280-284.	Deleted.
L285 "services"	Changed.
L286 "variations"	We changed to " seasonal and inter annual variation indexes ".
L295-296 Q_A and Q must have the same unit.	Correct, we changed to " km³/y "
L299 "storage is"	We changed to " The costs for additional short-term energy storage costs is are presented in equation (5) ".
L302 Add a comma after "\$".	Added.
L305 "E_Rwc and E_Rwoc"	Changed.
Figures & tables	
Fig. 1 Origin of the data plotted on this figure?	Added the text and reference below: "Figure adapted from 11". "C. Augustine et al., "Renewable Electricity Generation and Storage Technologies," in Renewable Electricity Futures Study , M. Hand, S. Baldwin, E. DeMeo, J. Reilly, T. Mai, D. Arent, G. Porro, M. Meshek, and D. Sandor, Eds. Golden: National Renewable Energy Laboratory, 2012.".
Zn-Br in the legend instead of VR.	Changed. You are a great reviewer! Thanks so much and sorry for the mistake.
L33 "seconds to year" (according to the figure).	This is great, thanks! If it is interesting to you, let me know if you would be interested in participate in my future papers.
Fig. 2 Font size is often too tiny.	True, increased the font size.
Fig. 2b c frame "Strahler"	Changed.
Fig. 2d What does "Dam Height = 50" mean? Unit?	Added unit. It looks for Dams higher than 50 meters.
Fig. 2g Add an arrow back in case of constant river flow from hydrological analysis?	As mentioned previously, this step does not discard a project. It just increases the final energy storage, if there is not enough water available to full the reservoir.
Fig. 2h Units in legend?	Added.
Fig. 3 Font size is often too tiny.	Increased the fonts.
Fig. 3a Turbine	Changed.
Fig. 3b US\$	Changed.
Energy storage costs with cascade.	Changed.
Fig. 3c Change for geographic coordinates.	Changed.
Reservoir is difficult to see but appears blue and not purple.	Increased reservoir and changed both reservoirs to purple.
Fig. 3d Units in legend?	Units added.
Fig. 4 Font size is too tiny.	I reshuffled the figure and increased the font.
Australia and Europe lines are the same colour (a b c d right side).	Changed.
Fig. 4e "Percentage of the reservoir that is filled with the tributary river inflow" (caption). y-axis legend "Tributary river flow percentage (%)" is therefore ambiguous.	Changed.
Supplementary Fig. 2	
L53 "extracts"	Changed.
L55 "withdrawals"	Changed.

Supplementary Figs. 3 and 4	
Add legend & coordinates.	The legend is in the Figure caption. We decided to delete Supplementary Figure 4 because the coordinates are too difficult to add.
Supplementary Fig. 3 is not entirely clear.	Changed.
Supplementary Tables (L line, C column)	
Supplementary Table 1	
L6C2 "consuming more electricity in the summer for cooling"	Changed.
No link between island electricity generation in L10C1 and costs in L10C2.	Changed.
Supplementary Table 2	
C6 Reference numbers are inappropriate, please check and correct (+3).	Thanks for spotting this.
Supplementary Table 3	
Erroneous "Links in the paper": starting from L2, should be Fig. 2a / Fig. 2b / Fig. 2d / Fig. 2e / Fig. 2f // Supplementary Tables 4 and 7 / ...	Changed.
L3C2 "This stage looks for a river"	Changed.
L6C2 No commas before and after "which side of the dam".	Changed.
L7C2 "within the reservoir"?	Changed.
L8C2 "This section calculates"	Changed.
Supplementary Table 4	
L14C1 S_v from Eq. 1.	Added.
L15C1 I_v from Eq. 2.	Added.
Same in Supplementary Table 5 L1C7 & L1C8.	You probably meant Supplementary Table 6.
L15C1 "variability" or "variation"	Changed.
Supplementary Table 5	
Please specify "in Tibet, China" in the table caption.	Changed.
L1C4, L1C6, L1C8 Add "cost".	Changed.
Supplementary Table 6	
Please specify "in Tibet, China" in the table caption.	Added.
Supplementary Table 7	
L8C2 " The hydrological data is in 0.1° resolution" (Supplementary Table 2, L4C3)	Changed.
"much lower than the 15 sec topography resolution"	Changed.
L15C1 "Open-loop SPS"	Changed.
L15C2 "Closed-loop SPS require two large reservoirs"	Changed.
"Additionally, there would be no hydropower plants in cascade that increase energy storage without additional costs."	Changed.
L16C2 "from the tributary river"	Changed.
L18C2 " The needs for energy and water storage are"	Changed.
References	
The references section could have been checked more carefully for accuracy before submission and must be	We am sorry, we thought that Mendeley would make sure that the reference were adjusted according to the Nature

improved: there are duplicates, second-hand references, erroneously numbered references (Supplementary Tables 1 & 2) and not up-to-date references. Details below.	Communications format.
L21-24 Rehman et al. (2014) provide an extensive review of pumped-storage systems. Rehman S., Al-Hadhrami L.M., Alam M.M. Pumped hydro energy storage system: A technological review. Renewable and Sustainable Energy Reviews 44, 586–598 (2015).	Added reference
L329-330 2. International Renewable Energy Agency (IRENA). Renewable power generation costs in 2014 . (2015).	Changed
IRENA 2019 report was recently published (May 2019) and could be of interest for this study. IRENA 2018 report provides an estimation of global installed capacity for year 2017 but its source is the DOE Global Energy Storage Database (http://www.energystorageexchange.org/). See also IRENA renewable energy statistics (July 2019).	Both added, we also added the reference below: “International Hydropower Association. Pumped Storage Tracking Tool. (2019). Available at: https://www.hydropower.org/hydropower-pumped-storage-tool”.
L331-332 This is a second-hand reference. Source is again the DOE (see previous comment).	Removed the reference.
L333 4. International Electrotechnical Commission. Electrical Energy Storage: White Paper .	Corrected
L366-367 18. Stocks, M. et al. A global atlas of pumped hydro energy storage (2019) . Available at: https://nationalmap.gov.au/renewables/#share=s-oDPMo1jDBBtwBNhD .	Changed
L368-369 19. Gimeno-Gutiérrez, M. & Lacal-Arántegui, R. Assessment of the European potential for pumped hydropower energy storage . JRC Scientific and Policy Reports (2013).	Added.
L370-371 20. Lacal-Arántegui, R., Fitzgerald N. & Leahy, N. Pumped-hydro energy storage: potential for transformation from single dams . JRC Scientific and Technical Reports (2012).	Added.
L368-369 19. and L370-371 20. could be replaced by Gimeno-Gutiérrez M., Lacal-Arántegui R. Assessment of the European potential for pumped hydropower energy storage based on two existing reservoirs . Renewable Energy 75, 856-868 (2015).	Replaced reference.
L390 28. Rognlien, L. Pumped Storage Development in Øvre Otra, Norway . MSc Thesis, Norwegian University of Science and Technology (2012).	Corrected.
L391-392 29. and L393 30. are the same.	Deleted.
Cost base for hydropower plants (With a generating capacity of more than 10,000 kW). Published by: Norwegian Water Resources and Energy Directorate. Editor: Jan Slappgård. Authors: SWECO Norge AS.	Changed.
L399 33. Inage, S-I	Corrected.
L406-408 Please delete the reference to this website.	Removed.
L409-410 37. International Energy Agency. Energy Technology Perspectives: scenarios & Strategies to 2050 . (2008).	Corrected the reference.

L414 Ref. 39. is redundant with the previous one (Ref. 38).	Deleted Ref 39.
L421-422 Where is Ref. 42 cited?	Deleted the reference.
Supplementary Table 1 (L line, C column):	
L3C1 Ref. 24 not appropriate here.	The order of the reference in the Supplementary Information was not correct. It is correct now.
L4C1 Ref. 31 not appropriate here (but in L3C1).	The order of the reference in the Supplementary Information was not correct. It is correct now.
L5C1 Ref. 34 focuses on a case study for the United States, not on high latitudes	The order of the reference in the Supplementary Information was not correct. It is correct now.
L6C2 Ref. 35 not appropriate here (but in L4C1).	The order of the reference in the Supplementary Information was not correct. It is correct now.
L8C1 Ref. 36 is not related at all to low energy security; ref. 37 not appropriate here (but in L10C2).	The order of the reference in the Supplementary Information was not correct. It is correct now.
L10C1 Ref. 40 & 41 (Canary Islands) instead of 38 & 39?	The order of the reference in the Supplementary Information was not correct. It is correct now.

9

10

11 **Reply to Reviewer #2:**

12

Reviewer #2 comments	Reply to Reviewer #2 comments
The strength of the paper is the global analysis for adequate sites for seasonal pump storage projects. The logic is to find favorable terrain conditions for the construction of a dam to store water (upper reservoir) nearby a river with enough water flows for the seasonal operation. The water is pumped from the river into the upper reservoir “parallel” to the river system, at a higher elevation in periods of low demand or favorable hydrology. It is then used for producing electricity in periods of high demand or insufficient water flows in the river. This approach has several alleged advantages with respect to the most common alternative of damming the river, such as the reduction of the socioenvironmental impact due to the smaller flooded areas, the fact that the river connectivity is unaffected and the possibility to supply fresh water to nearby communities.	Dear Dr. Rafael Kelman, The authors would like to thank you for pressure time improving the quality of the paper. We are very grateful for your contributions, hope that you could take something back from reading the paper.
The paper expands a previous work (reference 17) that focused on closed PHS systems that also investigated promising sites for PHS on a global scale. The main difference, however, is the configuration of the PHS system. Reference 17 investigates closed system, where both lower and upper reservoirs are created by dams (sometimes many dams are built to avoid points of “leakage” on the terrain) whereas the present paper focuses, as mentioned, siting one dam parallel to a river.	We added your valuable contribution to the text below in line “Open-Loop SPHS” in Supplementary Table 7 “Reference 17 presents work focused on closed-loop PHS systems on a global scale, where both lower and upper reservoirs are created by dams (sometimes many dams are built to avoid points of “leakage” on the terrain) whereas the model present in this paper focuses on siting one dam parallel to a main river”.
The structure of paper is not ideal. I spent a long time trying to understand the workflow of Fig.2. I was only able to decipher it could after reading section "Methods", which comes much later in the paper. It is also odd that "Discussion" precedes “Methods”.	We are sorry to hear that the paper structure is not ideal, we also agree with you. However, all Nature publications has the methods section after discussion, including Nature Communication. We would like to kindly request that you reconsider your demand as Nature Communication will asks

Work needs to be done to reorder the sections for the sake of clarity.	us to place the methods section after the discussion section.
Another important remark is that most assumptions, parameters and formulas are either not explained or insufficiently justified.	We have added the assumption and explanation to the model design decisions to all possible aspects that we could think about in the model, particularly the ones that you and the other reviewer requested.
References could be made when possible. To name a few examples, why is the maximum volume of the reservoir equal to 11% of the annual river flow? (it is evident that this is the cap from formula 3, but the rationale of this formula should be mentioned).	The authors created a methodology to estimate the extraction of water from the river as they could not find in the literature a methodology that could be effectively applied for SPHS plants in a global scale. We added two lines named “Comparison between storage capacity and cost” and “River water extraction” in Supplementary Table 7 to describe the methodology. “The comparison between the storage capacity and its cost is divided into the three sections below: (i) if the storage capacity is smaller than the water available to be stored, the costs for water and energy storage consists of the costs for the construction of the project in US\$ divided by the amount of water or energy that the project stored throughout the operation of the plant in km³ and MWh, respectively. The costs and services that happen throughout the operation of the plant is brought to today’s values with a discount factor of 18.2. (ii) if the storage capacity of the reservoir is smaller than the yearly amount of water available to store times two. Any additional storage capacity that exceeds QA will only reduce the cost of the water and energy storage costs by half when compared to QA. This is because the chances that the section of the reservoir destined to store energy seasonally has a higher change to fill up than the section destined to store energy inter-annually. (iii) if the storage capacity is bigger than twice the available yearly water availability, WS is fixed and equal to $1.5QA$. That means that any increase in storage capacity that surpasses twice the value of the yearly water availability will not contribute to lower the cost of water and energy storage. Note that the energy lost with the storage process is not included in the costs of the project and should be included when designing the plant. The energy efficiency assumed for optimizing the costs of the tunnels is 70% (a conservative value). This cost was not included because the costs of electricity and regulations for the operation of storage plants vary considerably from country to country.” “The authors created a methodology to estimate the extraction of water from the river as they could not find in the literature a methodology that could be effectively applied to SPHS plants, that (i) took into account the seasonal and inter-annual variation in a simple and comprehensive manner, (ii) could be applied on a global scale, (iii) guarantee that there would be enough water available to be pumped to the upper reservoir, (iv) to estimate a conservative upper reservoir capacity, which could fill up with the limited pumping capacity, (v) increase the viability of the construction of a SPHS plant, (vi) reduce the risks that the SPHS plant would not have enough water to operate. Another important aspect that was considered by the authors were the

	impacts of the SPHS plant in the basin hydrology. We compared three possible scenarios for the extraction of water from the river with high extraction (around 50% of the annual river flow), medium (30%) and low (10%). This design decision depends on policies developed by the stakeholders involved in the water management of a basin. Given that the model methodology was to be implemented on a global scale and that it would be complex and controversial to vary this assumption from basin to basin, we decided to be conservative in the amount of water that could be extracted from the river and selected the low extraction scenario. The final extraction is not 10%, but 11% due to the approach that the methodology proposed includes seasonal and inter-annual variation indexes in the analysis. For example, if the seasonal and inter-annual variation indexes are equal to 1, then the Equation 3 turns out to be $QA = Q \times 0.1 \times 1.1 = 0.11$, hence the limit of 11% for water extraction from the river.”.
Why is the maximum dam length 7.2 km?	We added a line named “Model parameters limitations” in Supplementary Table 7 to describe the parameter selection. “The model was designed to have the least possible restrictions. However, the model was taking too long to converge. Thus, we restricted the value of some parameters to reduce computational time. The run presented in this paper took one month to converge. The main restrictions are the (i) length of the dams of 7.2 km, (ii) size of the reservoirs of 1.620 km², (iii) dam heights of 50 to 250 m, and (iv) tunnel length or 3 to 30 km. The dam length and reservoir area were set high with the expectation that they would not result in viable projects, that is cheaper than 0.2, US\$/m³ for water storage and 50 US\$/MWh for energy storage. It turns out that there are viable projects with dams with 7,2 km, but no viable projects with areas higher than 1,620 km². If we would run the model again, we would increase the dam length to 14,4 km”.
Formulas (4) that present the calculation of economic performance of the seasonal pumped-storage (SPS) systems deserve further explanation, especially variable WS.	We added a line named “Comparison between storage capacity and cost” in Supplementary Table 7 to describe Equation 4.
A web-based GIS interface with the results of this analysis would greatly strengthen the findings of this paper. It could also allow for the download of the processed results, as in the case of reference 18. This should not be a difficult task, considering the several possible open GIS platforms available.	We agree that a web-based GIS interface that present the results would be a great approach to increase the impact of the paper, similarly to the AREMI project by the Australian Government. We will consider this request however. In this GIS implementation we would be able to add environmentally protected areas, population, biodiversity, transmission lines and etc. So that the user could see the impact of these restriction on the viability of the projects. As described in the paper submission request. We are will share any data resulted from this research freely up on request. We would be very happy if other institutions are interested in presenting the results of this work in a web-based GIS interface, as long as they reference this paper. The benefit of publishing in Nature Communications is that the paper will

The work has not evaluated the impacts of the dams with respect to the surrounding area because the work is based on topography only. If a dam floods a city, for example, it will not be discarded, despite being infeasible in real life. It is not clear many projects that have been screened would "survive" filters that would include other layers of information. The author recognizes this needs to be incorporated in the analysis in a future work.	be open source. This is a relevant issue, which we also had discussed thoroughly. We decided to study the potential for the technology because this is the first study that looks at the potential for SPHS in a global scale around the world. Including layers with protected areas, biodiversity, population, would not be a problem for the model. The issue is to analyze the impact of these restrictions in each country, for example, what is the minimum number of people required to cancel a SPHS project, or what is the costs for relocation for each individual in each of the countries under analysis. These assumption would be interesting to be worked in more details in future publication or in regional studies. Another case is the distance to transmission lines. There might be locations in the world that there is not human occupation due to the lack of water management solutions. These SPHS plants might contribute to the creation of new areas where civilization could potential explore socially and economically. However, if we added transmission line costs, we would restrict the projects where there is already civilization. With this in mind we decided to leave the analysis with the least restriction possible. The reader could then add the restrictions themselves.
Overall, I believe there is merit in the work, but it needs a thorough review before accepted.	We agreed that the paper has improved substantially with your contributions and appreciate your attention to detail.
I have also made comments and suggestions in the body of paper (Word file) for consideration.	We replied to your comments in the body of the Word file also.

REVIEWERS' COMMENTS:

Reviewer #1 (Remarks to the Author):

I thank the authors for paying careful attention to my earlier comments and suggestions, answering each of the observations in an appropriate manner and revising their manuscript accordingly. I found the revised version clear, particularly as regards the description of the model in Supplementary Tables, which helped clarify its strengths and limitations. I only have a few more edits before publication, which are detailed below.

Anne Jost

L20 "distributed with mountainous regions **d**emonstrating significantly more **potential**."

L28 "constitute **s**"

L61-62 "does not include cost analysis"

As mentioned earlier, these authors did not provide a detailed cost analysis but ranked nonetheless the identified sites according to an approximate cost model.

L62-63 "We have not included **these** closed loop **sites** because *they* are designed"

L65-66 "these are regional models **and** also do not include costs."

L107-108 "to reduce the impact **of** the SPHS plant in the river flow" ?

Fig. 3b "with casc**a**"

Fig. 3 "Example of **energy storage** cost variation"

L214 "distribution across"

L222-224 "This complementarity is usually the case in mid/high latitude countries, where during the summer river flow is higher due to ice melting and energy demand is lower compared to the winter."

Ice melting is not common at mid-latitudes outside mountain areas.

L236 "already tr**ied**"

L277 "and with a maximum length of 7.2 km"

Add comment in response to Reviewer#2 in Supplementary Table 3 (L4).

L285 "using the equations present**ed** in the reference"

L291 "and result**s** in a small impact to the river."

L331-332 "C_Ewc is the cost of long-term energy storage including the cascade in US\$/MWh"
To be cited L328 before E_Ewoc.

References

35. SWECO Norge AS. Cost base for hydropower plants (With a generating capacity of more than 10,000 kW). Published by: Norwegian Water Resources and Energy Directorate. Editor: Jan Slapgård. (2012).

Supplementary Tables (L line, C column)

Supplementary Table 1

L16C2 "Storing water in a SPHS reservoir parallel to the river, allows" No comma after "river".

"In these low water availability periods,"

"Water will be pumped into the river when the river flow rate"

"If the SPHS is still required to provide short term energy storage,"

L18C2 "The improvement in water management resulting from a SPHS plant would reduce the chances that a waterway runs out of water."

Supplementary Table 2

L4C4 & L5C4 "SPHS plant"

L5C4 "This gave a good preliminary estimate of the final costs."

"The selection of the turbine, also depends" No comma after "turbine".

Supplementary Table 3

Remove the blank line (L10)

Supplementary Table 4

L14C7 "0.f76"?

Supplementary Table 7

L4C2 "the costs of appropriate social and environmental measures in the construction of the SPHS reservoir are included in the project cost."

L6C2 "Other reasons why a reduced flow seasonality ... are due to environmental restrictions, river mouth and coastal sedimentation and other aspects." Missing words.

L7C2 Sequence of tenses in the first sentence.

"Another important aspect that was considered by the authors was the impacts of the SPHS plant in the basin hydrology."

"due to the approach that the methodology proposed includes" Rephrase.

L8C2 "cost assumptions [...] vary"

"for the Norwegian Krone to US\$"

L8C2 "the discount factor is 18.4" & L9C2 "with a discount factor of 18.2" Why two different values?

L9C2 Mention Equation 4.

"Any additional storage capacity that exceeds Q_A will only reduce the water and energy storage costs by half when compared to Q_A ." Rephrase.

Water storage cost C_w is reduced by half if $W_R = 2Q_A$ but over half if $W_R < 2Q_A$. Energy storage costs increase.

"This is because the section of the reservoir destined to store energy seasonally has a higher **chance** to fill up than the section destined to store energy inter-annually."

"That means that any increase in storage capacity that surpasses twice the value of the yearly water availability will not contribute to lower the cost of water storage."

"This cost was not included" Which cost?

L13C2 "There might be some uncertainties **originating from** the hydrological data." There *are* uncertainties in simulated river flows.

"Assuming that there are uncertainties associated with the PCR-GLOBWB global hydrological model, the error of this methodology is small." Uncertainties would not cancel each other out.

But in the end, the model may not be very sensitive to the estimated water availability?

L14C2 "The **cost** for tunnel excavation assumes an average value and does not include"

L16C2 Computing time on which type of computer?

"A resolution of 15 sec**,** is equivalent" No comma after "sec".

"because the flow variations per pixel within a river **are a** small fraction of the total river flow."

L23C2 "They require two **large** reservoirs and only store energy."

"building two **large** dams and reservoirs"

See Reviewer#2's comment#11 "Not necessarily."

L24C2 "it increases the possibility of increasing" Rephrase.

L25C2 "in some SPHS plants" Cf. L177 "Three [...] over more than 1,000"

L29C2 "The inclusion of lakes to the **model**"

L30C2 "tunnel length **of** 3 to 30 km"

"The **maximum** dam length and reservoir area were set high with the expectation that they **[and values over and above these maxima]** would not result in viable projects, that **are** cheaper" Ambiguous wording.

"viable projects with dams with 7**.**2 km"

"increase the dam length to 14**.**4 km"

L34C2 "it is possible that batteries will become cheaper **than** PHS plants"

L36 There was already a "Restriction zones" item L28: combine both?

L37C2 "For example^{22,26} also used SRTM topographic data to estimate the volume of **their** reservoirs and **validated** their reservoir storage capacity with a sample of reservoirs or lakes with known volumes, and **showed** that the error in estimating the water storage **was** small."

Supplementary Fig. 3

At least mention in the figure caption the topographic background (even better with a numbered colour scale).

Reviewer #2 (Remarks to the Author):

There have been considerable improvements in the paper. It is much clearer with this new structure. The rationale for the selection of some of the parameters and assumptions could be improved. I think this remark from my earlier review was not completely addressed. Overall, I am satisfied with the updated version of the paper. I did make some minor suggestions in the writing (including a lack of space to separate two words in the title!) (Please also see the attachment)

1
2
3
4
5
6
7
8

Response to Reviewers

Global resource potential of seasonal pumped hydropower storage for energy and water storage

Reply to Reviewer #1 comments:

Reviewer #1 comments	Reply to Reviewer #1 comments
I thank the authors for paying careful attention to my earlier comments and suggestions, answering each of the observations in an appropriate manner and revising their manuscript accordingly. I found the revised version clear, particularly as regards the description of the model in Supplementary Tables, which helped clarify its strengths and limitations. I only have a few more edits before publication, which are detailed below. Anne Jost	Dear Dr. Anne Jost, thanks a lot for your contributions. They have substantially improved the paper.
General comments	
L20 "distributed with mountainous regions demonstrating significantly more potential."	Changed.
L28 "constitutes"	Changed.
L61-62 "does not include cost analysis". As mentioned earlier, these authors did not provide a detailed cost analysis but ranked nonetheless the identified sites according to an approximate cost model.	We changed to "do not include detailed cost analysis".
L62-63 "We have not included these closed loop sites because they are designed"	Changed.
L65-66 "these are regional models and also do not include costs."	Changed.
L107-108 "to reduce the impact of the SPHS plant in the river flow"?	Changed.
Fig. 3b "with cascade"	Changed.
Fig. 3 "Example of energy storage cost variation"	Changed.
L214 "distribution across"	Changed.
L222-224 "This complementarity is usually the case in mid/high latitude countries, where during the summer river flow is higher due to ice melting and energy demand is lower compared to the winter." Ice melting is not common at mid-latitudes outside mountain areas.	Thanks, removed mid-latitudes.
L236 "already tried"	Changed.
L277 "and with a maximum length of 7.2 km" Add comment in response to Reviewer#2 in Supplementary Table 3 (L4).	Added Supplementary Table 3 (L4).
L285 "using the equations presented in the reference"	Changed.
L291 "and results in a small impact to the river."	Changed.
L331-332 "C_Ewc is the cost of long-term energy storage including the cascade in US\$/MWh" To be cited L328 before E_Ewoc.	Changed.
References	
35. SWECO Norge AS. Cost base for hydropower plants	Changed.

(With a generating capacity of more than 10,000 kW). Published by: Norwegian Water Resources and Energy Directorate. Editor: Jan Slapgård. (2012).	
Supplementary Tables	
Supplementary Table 1. L16C2 "Storing water in a SPHS reservoir parallel to the river, allows" No comma after "river".	Changed.
"In these low water availability periods,"	Changed.
"Water will be pumped into the river when the river flow rate"	Changed.
"If the SPHS is still required to provide short term energy storage,"	Changed.
L18C2 "The improvement in water management resulting from a SPHS plant would reduce the chances that a waterway runs out of water."	Changed.
Supplementary Table 2. L4C4 & L5C4 "SPHS plant".	Changed.
L5C4 "This gave a good preliminary estimate of the final costs."	Changed.
"The selection of the turbine, also depends" No comma after "turbine".	Changed.
Supplementary Table 3. Remove the blank line (L10)	Removed.
Supplementary Table 4. L14C7 "0.f76"?	Deleted "f".
Supplementary Table 7. L4C2 "the costs of appropriate social and environmental measures in the construction of the SPHS reservoir are included in the project cost."	Changed.
L6C2 "Other reasons why a reduced flow seasonality ... are due to environmental restrictions, river mouth and coastal sedimentation and other aspects." Missing words.	Added "is not appropriate".
L7C2 Sequence of tenses in the first sentence. "Another important aspect that was considered by the authors was the impacts of the SPHS plant in the basin hydrology."	Changed
"due to the approach that the methodology proposed includes" Rephrase.	Rephrased.
L8C2 "cost assumptions [...] vary".	Changed.
"for the Norwegian Krone to US\$"	Changed.
L8C2 "the discount factor is 18.4" & L9C2 "with a discount factor of 18.2" Why two different values?	Thanks for spotting this. The correct value is 18.2.
L9C2 Mention Equation 4. "Any additional storage capacity that exceeds Q_A will only reduce the water and energy storage costs by half when compared to Q_A ."	Changed and added Equation 4.
Rephrase. Water storage cost C_w is reduced by half if $W_R=2Q_A$ but over half if $W_R<2Q_A$. Energy storage costs increase.	We could not understand this comment.
"This is because the section of the reservoir destined to store energy seasonally has a higher chance to fill up than the section destined to store energy inter-annually"	Changed.
"That means that any increase in storage capacity that surpasses twice the value of the yearly water availability will not contribute to lower the cost of water storage."	Changed.

"This cost was not included" Which cost?	Changed to: The electricity cost for pumping was not included because the costs of electricity and regulations for the operation of storage plants vary considerably from country to country.
L13C2 "There might be some uncertainties originating from the hydrological data." There are uncertainties in simulated river flows.	Changed.
"Assuming that there are uncertainties associated with the PCR-GLOBWB global hydrological model, the error of this methodology is small." Uncertainties would not cancel each other out. But in the end, the model may not be very sensitive to the estimated water availability?	Deleted the phrase and replaced with the phrase below: "There are not many reservoirs that can be filled completely with the water available for extraction. In other words, there is usually more water available than storage capacity. Thus, the water availability is not the main limitation for SPS potential, but the topography."
L14C2 "The cost for tunnel excavation assumes an average value and does not include"	Changed.
L16C2 Computing time on which type of computer?	Added more details on the machine "The run presented in this paper took one month to converge with a desktop with a processor Intel Core i7-6700, 3.4 GHz x 2".
"A resolution of 15 sec, is equivalent" No comma after "sec".	Changed.
"because the flow variations per pixel within a river are a small fraction of the total river flow."	Changed.
L23C2 "They require two large reservoirs and only store energy." "building two large dams and reservoirs" See Reviewer#2's comment#11 "Not necessarily."	Changed and added "usually".
L24C2 "it increases the possibility of increasing" Rephrase.	Changed to: "This arrangement is interesting because it increases the total generation head of the plant".
L25C2 "in some SPS plants" Cf. L177 "Three [...] over more than 1,000".	Changed to: "in three over more than a thousand SPS plants".
L29C2 "The inclusion of lakes to the model"	Changed.
L30C2 "tunnel length of 3 to 30 km".	Changed.
"The maximum dam length and reservoir area were set high with the expectation that they [and values over and above these maxima] would not result in viable projects, that are cheaper".	Changed.
Ambiguous wording. "viable projects with dams with 7.2 km".	Added "Dam length higher than".
"increase the dam length to 14.4 km".	Changed.
L34C2 "it is possible that batteries will become cheaper than PHS plants"	Changed.
L36 There was already a "Restriction zones" item L28: combine both?	Merged both columns.
L37C2 "For example ^{22,26} also used SRTM topographic data to estimate the volume of their reservoirs and validated their reservoir storage capacity with a sample of reservoirs or lakes with known volumes, and showed that the error in estimating the water storage was small."	Changed
Supplementary Fig. 3. At least mention in the figure caption the topographic background (even better with a numbered colour scale).	Added the topographic background.

10 Reply to Reviewer #2 comments:

11

Reviewer #2 comments	Reply to Reviewer #2 comments
There have been considerable improvements in the paper. It is much clearer with this new structure.	Dear Dr. Kelman, Thanks a lot for your positive feedback and for continuing improving the manuscript.
The rationale for the selection of some of the parameters and assumptions could be improved. I think this remark from my earlier review was not completely addressed.	We agree that there are still some clarifications to be made, however we have clarified the most important aspects and the aspects that the reviews explicitly requested. In a future paper we will document better all decision so that we can report all decision in the paper.
Overall, I am satisfied with the updated version of the paper.	Thanks!
I did make some minor suggestions in the writing (including a lack of space to separate two words in the title!) (Please also see the attachment).	Thanks for your further contribution. We added your suggestions to the paper.

12

13 Reply to Editor comments:

14

Editor comments	Reply to Editor comments
The final version of any Supplementary Information (figures, tables, notes etc) in one PDF file. Please add a cover page to the Supplementary Information PDF, including the title of the manuscript and the first author's surname in the format 'Smith et al.'	We added the title of the paper and the first author surname to the Supplementary Information cover page.
Please submit movies, audio files and data sets as separate files. See http://www.nature.com/ncomms/submit/how-to-submit#Supplementary-information for acceptable file formats/sizes.	There are no movies in this paper.
Please rearrange SI into the following order: Supplementary Figures (Supplementary Figure1,2,3..); Supplementary Tables (Supplementary Table 1,2,3..); Supplementary Notes (Supplementary Note1,2,3..); Supplementary Discussion; Supplementary References. Supplementary Figures/Tables can only contain figures/tables without text, while Supplementary Notes and Supplementary Discussion can only contain text without figures or tables.	The paper is following these rules.
Please ensure all Supplementary Figures/Tables/Notes have been cited in order in the main article.	In order to cite the Supplementary Tables in order, we changed the Supplementary Tables 1, 7, 2, 3, 4, 5, 6, to 1, 2, 3, 4, 5, 6, 7, respectively. In order to cite the Supplementary Figures in order, we changed the Supplementary Figures 2, 1, 3 to 1, 2, 3, respectively.

15

16 Reply to Editor requests:

17

Editor requests	Reply to Editor requests
* We encourage increased transparency in peer review by publishing the reviewer comments and author rebuttal letters of our research articles, if the authors agree. Such peer review material is made available as a supplementary peer review file. Please state in the cover letter 'I wish to participate in transparent peer review' if you want to opt in, or 'I do not wish to participate in transparent peer review' if you don't. Failure to state your preference will results	I wish to participate in transparent peer review.

in delays in accepting your paper for publication. Please note: we allow redactions to authors' rebuttal and reviewer comments in the interest of confidentiality. If you are concerned about the release of confidential data, please let us know specifically what information you would like to have removed. Please note that we cannot incorporate redactions for any other reasons. Reviewer names will be published in the peer review files if the reviewer signed the comments to authors, or if reviewers explicitly agree to release their name. For more information, please refer to our FAQ page at: https://www.nature.com/documents/ncomms-transparent-peer-review.pdf	
* Please ensure that an updated editorial policy checklist that verifies compliance with all required editorial policies is completed and uploaded with the revised article. All points on the policy checklist must be addressed; if needed, please revise your manuscript in response to these points. Please note that this form is a dynamic 'smart pdf' and must therefore be downloaded and completed in Adobe Reader, instead of opening it in a web browser. Editorial policy checklist: https://www.nature.com/documents/nr-editorial-policy-checklist.pdf	We filed up the check list and attached to the submission.
* Your manuscript should comply with our policies and format requirements, detailed in our checklist for authors at: https://www.nature.com/documents/ncomms-manuscript-checklist.pdf	The paper complies with the policies and format requirements.
* Please also review the changes in the attached copy of your manuscript, which has been edited for style, and address the comments and queries I have added. If using Word, please use the 'track changes' feature to make the process of accepting your manuscript more efficient.	We have addressed your comments using 'track changes'.
* Data availability statements and data citations policy: All Nature Communications manuscripts must include a section titled "Data Availability" as a separate section after the Methods section but before the References. For more information on this policy, and a list of examples, please see https://www.nature.com/documents/nr-data-availability-statements-data-citations.pdf. In particular, the Data availability statement should include:  - Accession codes for deposited data. - Other unique identifiers (such as DOIs and hyperlinks for any other datasets). - At a minimum, a statement confirming that all relevant data are available from the authors. - If applicable, a statement regarding data available with restrictions - If a dataset has a Digital Object Identifier (DOI) as its unique identifier, we strongly encourage including this in the Reference list and citing the dataset in the Data Availability Statement. - If a source data file is provided, please add a reference to this in the data availability statement. For example:  - "The source data underlying Figs 1a, 2a–d, 6d, h and 7c and Supplementary Figs 1a and 5d are provided as a Source Data file." 	We added a data availability statement to the paper.
*DATA SOURCES: Nature Research policies strongly encourage deposition of research data in public repositories and in some cases this is mandatory, and you may have been previously advised if that was the case. If you need help depositing and curating your research data (including raw and processed data, text, video, audio and images) you should consider:  - Contacting Springer Nature's Research Data Helpdesk for advice - Finding a suitable data repository for your data - Uploading your data to Springer Nature's Research Data Support service Research Data Support is an optional Springer Nature service. There are fees for using this service, however, if you receive funding from the Wellcome Trust or are affiliated to a Wellcome Centre you can use Research Data Support at no	The data from this paper is not deposited in a public repository. We will deliver the data upon request.

cost. See here for more information. Please provide a unique identifier for the data (for example a DOI or a permanent URL) in the data availability statement, if possible. If the repository does not provide identifiers, we encourage authors to supply the search terms that will return the data. For data that have been obtained from publicly available sources, please provide a URL and the specific data product name in the data availability statement. Data with a DOI should be included in the reference list and cited where relevant. Alternatively, include the data in the Supplementary Information. For datasets for which mandatory deposition is not required and the data can only be shared on request, please explain why in your Data Availability Statement and in your cover letter. Please refer to our data policies here: http://www.nature.com/authors/policies/availability.html	
* To ensure correct hyperlinking of the accession codes in your manuscript, please add the hyperlink or DOI in square brackets directly after the code throughout (for example, '5XRN [http://dx.doi.org/10.2210/pdb5XRN/pdb]', '1483958 [https://dx.doi.org/10.5517/ccdc.csd.cc11t5m6]', 'SRP109982 [https://www.ncbi.nlm.nih.gov/sra/?term=SRP109982]' or 'NQLW00000000 [https://www.ncbi.nlm.nih.gov/assembly/GCA_002312845.1/]').	We followed this standard for DOI statements. However, we did not add the DOI for all references.
* Please check whether your manuscript or Supplementary Information contain third-party images, such as figures from the literature, stock photos, clip art or commercial satellite and map data. We strongly discourage the use or adaptation of previously published images, but if this is unavoidable, please request the necessary rights documentation to re-use such material from the relevant copyright holders and return this to us when you submit your revised manuscript.	Fig. 1 has been adapted from 'Augustine, C. et al. Renewable Electricity Generation and Storage Technologies. in Renewable Electricity Futures Study (eds. Hand, M. et al.) (National Renewable Energy Laboratory, 2012)'. This figure can be reproduced in other publication if the reference above is included (https://www.nrel.gov/web-standards/legal.html). We added the reference above in the figure caption. All other figures were created by the authors.
* Please ensure that an updated reporting summary is completed and uploaded with the revised article. All points on the reporting summary must be addressed; if needed, please revise your manuscript in response to these points. Please note that this form is a dynamic 'smart pdf' and must therefore be downloaded and completed in Adobe Reader, instead of opening it in a web browser. Reporting summary: https://www.nature.com/documents/nr-reporting-summary.pdf	We filed up the reporting summary and attached to the submission.

* The source data file should, as a minimum, contain the raw data underlying all reported averages in graphs and charts, and uncropped versions of any gels or blots presented in the figures. To learn more about our motivation behind this policy, please see https://www.nature.com/articles/s41467-018-06012-8. Within the source data file, each figure or table (in the main manuscript and in the Supplementary Information) containing relevant data should be represented by a single sheet in an Excel document, or a single .txt file or other file type in a zipped folder. Blot and gel images should be pasted in and labelled with the relevant panel and identifying information such as the antibody used. We also encourage you to include any other types of raw data that may be appropriate. An example source data file is available demonstrating the correct format: https://www.nature.com/documents/ncomms-example-source-data.xlsx The file should be labelled 'Source Data', with the title and a brief description included in your cover letter, and should be mentioned in all relevant figure legends using the template text below: "Source data are provided as a Source Data file."	The data is provided upon request to the authors.
* Reporting and materials availability requirements for Earth sciences research: http://www.nature.com/authors/policies/availability.html#requirements	We have followed the paper standards.
* We are committed to ensuring clarity and avoiding ambiguity in the mathematics in our papers. Consequently, please carefully check the mathematical terms throughout your manuscript (including labels on figures and figure captions) to ensure that it conforms strictly to the following guidelines. In mathematical terms, scalar variables (e.g. x, V, χ) should be typeset in italic, whereas multi-letter variables should be formatted without italic. Constants (e.g. h, G, c) should be typeset in italics (the only exceptions being e, i, π, which should be typeset without italic) and vectors (such as r, the wavevector k, or the magnetic field vector B) should be typeset in bold without italics. In contrast, subscripts and superscripts should only be italicized if they too are variables or constants. Those that are labels (such as the 'c' in the critical temperature, T_c, the 'F' in the Fermi energy, E_F, or the 'crit' in the critical current, I_{crit}) should be typeset in roman. Please also ensure the same convention is followed in figure labels, axes, and such. Additionally, to avoid doubt, unit dimensions should be expressed using negative integers (e.g. $\text{kg m}^{-1} \text{s}^{-2}$ not kg/ms^2) or the word 'per'.	We changed the equations and units according to the journal standards.
* Your paper will be accompanied by a two-sentence editor's summary, of between 250-300 characters, when it is published on our homepage. Could you please approve the draft summary below or provide us with a suitably edited version. The potential of seasonal Pumped-Storage (SPS) plant to fulfil future energy storage requirements is not well understood. Here the authors show that SPS costs vary from 0.007 to 0.2 \$/m³ of water stored, 1.8 to 50 \$/MWh of energy stored and 0.37 to 0.6 \$/GW of installed power generation capacity.	Please see below some suggested changes to the editor's summary. "The potential of Seasonal Pumped Hydropower Storage (SPHS) plant to fulfil future energy storage requirements is not well understood. Here the authors show that SPHS costs vary from 0.007 to 0.2 \$ m⁻³ of water stored, 1.8 to 50 \$ MWh⁻¹ of energy stored and 0.37 to 0.6 \$ GW⁻¹ of installed power generation capacity".
* As part of our efforts to communicate our content to a wider audience, we endeavour to highlight papers published in Nature Communications on the journal's Twitter account (@NatureComms). If you would like us to mention authors, institutions or lab groups in these tweets, please provide the relevant twitter handles in your cover letter upon resubmission.	Please mention the Energy and Water Programs at IIASA.
If you opted into the journal hosting details of a preprint version of your manuscript via a link on our dedicated website (https://nature-research-underconsideration.nature.com), it will remain on this site while you are revising your	We have published a preprint version of the paper in the link below: https://eartharxiv.org/5s7bt/

manuscript, as we consider the file to remain active. Should you wish to remove these details, please email naturecommunications@nature.com indicating your manuscript number and the link on our website that was previously sent to you. Please see our pre-publicity policy at http://www.nature.com/authors/policies/confidentiality.html For more information, please refer to our FAQ page at https://nature-research-under-consideration.nature.com/posts/19641-frequently-asked-questions	Please feel free to maintain the link for as long as you feel appropriate.
In recognition of the time and expertise our reviewers provide to Nature Communications's editorial process, as of November, 2018, we formally acknowledge their contribution to the external peer review of articles published in the journal. All peer-reviewed content will carry an anonymous statement of peer reviewer acknowledgement, and for those reviewers who give their consent, we will publish their names alongside the published article. For more information, please refer to our FAQ page: https://www.nature.com/documents/ncomms-reviewer-information.pdf	Thanks for recognizing the reviewer's contributions. They have considerably improved the paper.
Open Access	
Nature Communications is a fully open access journal. Articles are made freely accessible on publication under a CC BY license (Creative Commons Attribution 4.0 International License). This license allows maximum dissemination and re-use of open access materials and is preferred by many research funding bodies. For further information about article processing charges, open access funding, and advice and support from Nature Research, please visit http://www.nature.com/ncomms/about/open-access	Thanks for this information.
Submission Information	
* A cover letter describing your response to our editorial requests.	We added a cover letter describing the author response to the editorial requests.
* A separate document detailing your point-by-point response to any issues raised by our referees (please include the referees' comments in this document).	This document details the point-by-point response to reviewers and editor comments.
* The final version of your text as a Word or TeX/LaTeX file, with any tables prepared using the Table menu in Word or the table environment in TeX/LaTeX and using the 'track changes' feature in Word.	We used the table menu in word and 'track changes'.
* The complete author list provided in the article file, which must match that given on our manuscript tracking system. The author list in the main article file will be used during typesetting of your article.	We have checked that the name of the authors in the manuscript are correct.
* Production-quality versions of all figures, supplied as separate files containing all panels. To ensure the swift processing of your paper please provide the highest quality, vector format, versions of your images (.ai, .eps, .psd) where available. Please see our brief guide to manuscript submission for further details on the figure formats we can accept. Text and labelling should be in a separate layer to enable editing during the production process. If vector files are not available then please supply the figures in whichever format they were compiled in and not saved as flat .jpeg or .TIFF files. Any chemical structures or schemes contained within figures should additionally be supplied as separate ChemDraw (.cdx) files. If your artwork contains any photographic images, please ensure these are at least 300 dpi. To ensure that your figures are accessible to colour-blind readers, we encourage you to use alternative colour schemes. For example, rainbow colour scales may be replaced by single-colour intensity scales or greyscale, and red/green image overlays may be replaced with magenta/green. For reference an example of R-	We have provided the figures separately in JPEG format.

script colour blindness palettes can be found here https://cran.r-project.org/web/packages/viridis/vignettes/intro-to-viridis.html . Another example for Python can be found here: http://matplotlib.org/cmocean/	
* The final version of any Supplementary Information (figures, tables, notes etc) in one PDF file. Please add a cover page to the Supplementary Information PDF, including the title of the manuscript and the first author's surname in the format 'Smith et al.' Please submit movies, audio files and data sets as separate files. See http://www.nature.com/ncomms/submit/how-to-submit#Supplementary-information for acceptable file formats/sizes.	We added the title of the manuscript and the first author surname to the Supplementary Information.
** Please note that Supplementary Information must be finalised prior to acceptance of the paper.	The Supplementary Information section has been finalized.
* If you wish, an interesting image (but not an illustration or schematic) for consideration as a 'Featured Image' on the Nature Communications homepage. Examples can be seen on our Facebook page: http://go.nature.com/PGPizM The file should be 1400x400 pixels in RGB format and should be uploaded as 'Related Manuscript File'. In addition to our home page, we may also use this image (with credit) in other journal-specific promotional material.	We uploaded a 'Featured Image' for this paper. The image comes from the site below. These images are free from copyright: https://unsplash.com/s/photos/kaprun
* A completed author checklist, uploaded as a Related Manuscript file type, available at: https://www.nature.com/documents/ncomms-manuscript-checklist.pdf	We uploaded the manuscript checklist.
* Completed and signed copies of our Multimedia License to Publish (LTP) for any Featured Image suggestions (please use one form for each image and give a scientific description of the image in the 'title' field; do not use "Featured Image" as a title): Multimedia Licence to Publish form.	We completed and signed the Multimedia License to Publish.
At acceptance, the corresponding author will be required to complete an Open Access Licence to Publish on behalf of all authors, declare that all required third party permissions have been obtained and provide billing information in order to pay the article-processing charge (APC) via credit card or invoice.	We will cover the costs for publication as soon as he have the invoice.
Please note that your paper cannot be sent for typesetting to our production team until we have received these pieces of information; therefore, please ensure that you have this information ready when submitting the final version of your manuscript.	We have submitted that the information requested by the editor.
Springer Nature encourages all authors and reviewers to adopt an Open Researcher and Contributor Identifier (ORCID). ORCID is a community-based initiative that provides an open, non-proprietary and transparent registry of unique identifiers to help disambiguate research contributions. All authors who link their ORCID to their account in our submission system will have their ORCID published on their articles. Please note that this is only possible if ORCIDs are linked prior to acceptance, that is, it is not possible to add ORCIDs at proof. Please ensure that all co-authors are aware that they can link their ORCIDs, so that it will display on this paper. If they so wish, they must do so before the paper is formally accepted. It will not be possible to add ORCIDs post-acceptance, e.g. at proof. To link an ORCID please follow these instructions: 1. From the home page of the MTS click on 'Modify my Springer Nature account' under 'General tasks' 2. In the 'Personal profile' tab, click on 'ORCID Create/link an Open Researcher Contributor ID (ORCID)'. This will re-direct you to the ORCID website. 3a. If you already have an ORCID account, enter your ORCID email and password and click on 'Authorize' to link your ORCID with your account on the MTS. 3b. If you don't yet have an ORCID account, you can easily create one by providing the required information and then clicking on 'Authorize'. This will link your newly created ORCID with your account on the MTS.	I have already linked my ORCID.

If you experience problems in linking your ORCID, please contact Platform Support Please use the following link to submit the above items: https://mts-ncomms.nature.com/cgi-bin/main.plex?el=A5S3BagR7B2JNzk6I1A9fdO4Wln2ZehcHITUU1aXB3IAZ	
** This url links to your confidential home page and associated information about manuscripts you may have submitted or be reviewing for us. If you wish to forward this email to co-authors, please delete the link to your homepage first ** We hope to hear from you within two weeks; please let us know if the process may take longer.	Thanks for letting me know.
Nature Research journals encourage authors to share their step-by-step experimental protocols on a protocol sharing platform of their choice. Nature Research's Protocol Exchange is a free-to-use and open resource for protocols; protocols deposited in Protocol Exchange are citable and can be linked from the published article. More details can be found at www.nature.com/protocolexchange/about.	We have not added a Research's Protocol Exchange to this submission.
** See Nature Research's author and referees' website at www.nature.com/authors for information about policies, services and author benefits	Thanks for sharing the link.
This email has been sent through the Springer Nature Tracking System NY-610A-NPG&MTS	Thanks for sharing the tracking system.
Confidentiality Statement	
This e-mail is confidential and subject to copyright. Any unauthorised use or disclosure of its contents is prohibited. If you have received this email in error please notify our Manuscript Tracking System Helpdesk team at http://platformsupport.nature.com . Details of the confidentiality and pre-publicity policy may be found here http://www.nature.com/authors/policies/confidentiality.html Privacy Policy Update Profile DISCLAIMER: This e-mail is confidential and should not be used by anyone who is not the original intended recipient. If you have received this e-mail in error please inform the sender and delete it from your mailbox or any other storage mechanism. Springer Nature Limited does not accept liability for any statements made which are clearly the sender's own and not expressly made on behalf of Springer Nature Ltd or one of their agents. Please note that Springer Nature Limited and their agents and affiliates do not accept any responsibility for viruses or malware that may be contained in this e-mail or its attachments and it is your responsibility to scan the e-mail and attachments (if any).	Thanks for sharing this confidential statement. We will follow your requests.